# Tight Margin-Based Generalization Bounds for Voting Classifiers over Finite Hypothesis Sets

Kasper Green Larsen [* 1]   Natascha Schalburg [* 1]

## Abstract

We prove the first margin-based generalization bound for voting classifiers, that is asymptotically tight in the tradeoff between the size of the hypothesis set, the margin, the fraction of training points with the given margin, the number of training samples and the failure probability.

## 1. Introduction

Ensemble learning is a powerful machine learning tool; it enables us to transform weak learners; hypothesis classes that are barely better than guessing, into learners with state-of-the-art performance.

In essence, ensemble methods take a set of base classifiers, weigh those classifiers according to performance on the training set and retrieve the final prediction by aggregating according to those weights. An important historical example is AdaBoost ((Freund & Schapire, 1997)), a type of voting classifier, which builds the ensemble classifier sequentially; new base classifiers are added to the ensemble to correct the mistakes of the current ensemble. AdaBoost was the first efficient and practical implementation of a boosting algorithm, and hence the relevance of ensemble learners is often attributed to AdaBoost.

Much theoretical research has been done to explain the impressive practical performance of AdaBoost and other ensemble methods. In the groundbreaking work of (Schapire et al., 1998), it was first demonstrated experimentally that the accuracy of the voting classifier produced by AdaBoost sometimes keeps improving even when training past the point of perfectly classifying the training data. This goes against the conventional wisdom that simple models generalize better. In an attempt to explain this phenomenon,

---

[*]Equal contribution   [1]Department of Computer Science, Aarhus University, Aarhus, Denmark. Correspondence to: Kasper Green Larsen <larsen@cs.au.dk>, Natascha Schalburg <n.schalburg@cs.au.dk>.

*Proceedings of the $43^{rd}$ International Conference on Machine Learning*, Seoul, South Korea. PMLR 306, 2026. Copyright 2026 by the author(s).

the authors first observed that the so-called margins of the produced voting classifier kept improving when training past the point of perfect training accuracy. This led the authors to introduce the currently most prominent framework for analyzing the generalization performance of voting classifiers. This framework is called margin theory, named after the notion of margins, which can be interpreted as how *confident* the classifier is in its prediction.

In this paper we consider voting classifiers, i.e. ensemble learners in the setting of binary classification over a domain $\mathcal{X}$ with labels $\{-1, +1\}$. Formally, we model a voting classifier $f$ as a convex combination of base classifiers $h \in \mathcal{H}$. The set of voting classifiers $\mathcal{C}(\mathcal{H})$ over a base hypothesis set $\mathcal{H}$ is defined as:

$$\mathcal{C}(\mathcal{H}) = \left\{ f = \sum_{h \in \mathcal{H}} a_h h \ \middle| \ \sum_{h \in \mathcal{H}} a_h = 1, \, a_h \geq 0 \ \forall h \in \mathcal{H} \right\},$$

where the constant $a_h$ is the weight describing the impact of $h$ in the voting classifier. A voting classifier $f \in \mathcal{C}(\mathcal{H})$ makes a prediction on a point $x \in \mathcal{X}$ by computing $\mathrm{sign}(f(x))$. For every voting classifier $f \in \mathcal{C}(\mathcal{H})$, the margin of a data point $(\mathbf{x}, \mathbf{y}) \in \mathcal{X} \times \{-1, +1\}$ is defined as

$$y f(x) = y \sum_{h \in \mathcal{H}} a_h h(x) = \sum_{\substack{h \in \mathcal{H} \\ h(x)=y}} a_h - \sum_{\substack{h \in \mathcal{H} \\ h(x) \neq y}} a_h.$$

The margin thus lies in $[-1, +1]$. If a voting classifier's prediction is wrong on a data point $(\mathbf{x}, \mathbf{y})$, $\mathbf{y}$ and $f(\mathbf{x})$ will have different signs, making the margin negative. For a voting classifier, the magnitude of the margin depends on the consensus among the base classifiers; when base classifiers disagree their weights $\alpha_h$ will partially cancel out, making the margin smaller. In a nutshell, margin theory now says that if a voting classifier has large (positive) margins on its training data, then it generalized well to new data.

When studying generalization bounds for voting classifiers, one makes the standard assumption that the training data $\mathbf{S}$ is obtained as $n$ i.i.d. samples from an unknown distribution $\mathcal{D}$ over $\mathcal{X} \times \{-1, +1\}$. The goal is to bound the probability of miss-classifying a new data point $(\mathbf{x}, \mathbf{y}) \sim \mathcal{D}$, i.e. to

show that $\mathcal{L}_\mathcal{D}(f) = \mathbb{P}_{(\mathbf{x},\mathbf{y})\sim\mathcal{D}}[\mathbf{y}f(\mathbf{x}) \leq 0]$ is small. In margin theory, the loss $\mathcal{L}_\mathcal{D}(f)$ is related to the margins on the training data. For any margin $0 < \theta \leq 1$, we define $\mathcal{L}^\theta_\mathbf{S}(f) = \mathbb{P}_{(\mathbf{x},\mathbf{y})\sim\mathbf{S}}[\mathbf{y}f(\mathbf{x}) \leq \theta] = \frac{1}{n}\sum_{i\in[n]} \mathbb{1}_{\{\mathbf{y}_i f(\mathbf{x}_i)\leq\theta\}}$ as the fraction of training points $(\mathbf{x},\mathbf{y}) \in \mathbf{S}$ where $f$ has a margin of at most $\theta$. Here the notation $(\mathbf{x},\mathbf{y}) \sim \mathbf{S}$ denotes a uniform random point from $\mathbf{S}$.

The first margin-based generalization bound for voting classifiers was proved by (Schapire et al., 1998). They showed that for any distribution $\mathcal{D}$ over $\mathcal{X} \times \{-1,+1\}$, any training set $\mathbf{S} \sim \mathcal{D}^n$ of size $n$, any finite set of base classifiers $\mathcal{H}$, any $0 < \delta < 1$ and any margin $0 < \theta \leq 1$, it holds with probability at least $1 - \delta$ (over the random choice of $\mathbf{S}$) that *every* voting classifier $f \in \mathcal{C}(\mathcal{H})$ satisfies

$$\mathcal{L}_\mathcal{D}(f) \leq \mathcal{L}^\theta_\mathbf{S}(f) + c\left(\sqrt{\frac{\ln(n)\ln|\mathcal{H}|}{\theta^2 n}} + \frac{\ln(e/\delta)}{n}\right), \quad (1)$$

where $c > 0$ is a universal constant. Since then, much research has gone into understanding the behavior and impact of margins on voting classifiers. We discuss many of these research directions below and for now focus on the main topic of this work, namely margin-based generalization bounds in the simplest setup where the base hypothesis set $\mathcal{H}$ is finite.

Following the seminal work of (Schapire et al., 1998), (Breiman, 1999) studied the special case where $\mathcal{L}^\theta_\mathbf{S}(f) = 0$, i.e. when voting classifiers $f$ have margins at least $\theta$ on all training points. There, he showed that the bound in (1) improves to

$$\mathcal{L}_\mathcal{D}(f) \leq c\left(\frac{\ln(n)\ln|\mathcal{H}|}{\theta^2 n} + \frac{\ln(e/\delta)}{n}\right),$$

for all voting classifiers $f$ with $\mathcal{L}^\theta_\mathbf{S}(f) = 0$. The current best upper bound on the generalization error, due to (Gao & Zhou, 2013), interpolates between two above bounds and gives that with probability $1 - \delta$, it holds for every voting classifier $f \in \mathcal{C}(\mathcal{H})$ that

$$\mathcal{L}_\mathcal{D}(f) \leq \mathcal{L}^\theta_\mathbf{S}(f)$$
$$+ c\cdot\sqrt{\mathcal{L}^\theta_\mathbf{S}(f)\left(\frac{\ln(n)\ln(|\mathcal{H}|)}{\theta^2 n} + \frac{\ln(e/\delta)}{n}\right)}$$
$$+ c\cdot\left(\frac{\ln(n)\ln(|\mathcal{H}|)}{\theta^2 n} + \frac{\ln(e/\delta)}{n}\right). \quad (2)$$

A natural question is whether the bound of (Gao & Zhou, 2013) is tight for finite $\mathcal{H}$. This question was studied in two subsequent works (Grønlund et al., 2019a; 2020) on generalization *lower bounds*. The latter of these gives the currently tightest lower bound, stating that for any cardinality $N$, parameters $1/N < \tau, \theta < c$, and number of samples $c^{-1}\theta^{-2}\ln(N) \leq n \leq 2^{N^c}$, for a small constant $c > 0$,

there exists a data distribution $\mathcal{D}$ over $\mathcal{X} \times \{-1,+1\}$ and a finite hypothesis class $\mathcal{H}$ with $|\mathcal{H}| = N$ such that with constant probability over a training set $\mathbf{S} \sim \mathcal{D}^n$, there is a voting classifier $f \in \mathcal{C}(\mathcal{H})$ such that $\mathcal{L}^\theta_\mathbf{S}(f) \leq \tau$ and

$$\mathcal{L}_\mathcal{D}(f) \geq \tau + c\cdot\sqrt{\tau\cdot\frac{\ln(e/\tau)\ln(N)}{\theta^2 n}}$$
$$+ c\cdot\frac{\ln(\theta^2 n/\ln N)\ln(N)}{\theta^2 n}$$
$$\geq \mathcal{L}^\theta_\mathbf{S}(f) + c\cdot\sqrt{\mathcal{L}^\theta_\mathbf{S}(f)\cdot\frac{\ln(e/\mathcal{L}^\theta_\mathbf{S}(f))\ln(|\mathcal{H}|)}{\theta^2 n}}$$
$$+ c\cdot\frac{\ln(\theta^2 n/\ln(|\mathcal{H}|))\ln(|\mathcal{H}|)}{\theta^2 n}. \quad (3)$$

Note that the parameter $\tau$ is used to yield a lower bound for the entire tradeoff of possible values of $\mathcal{L}^\theta_\mathbf{S}(f)$. Let us also mention that the $\delta$-dependency in (2) can be shown to be tight following classic work, e.g. see Chapter 14 in (Devroye et al., 1996). Finally, the original work of (Grønlund et al., 2020) states the lower bound only for $(\theta^{-2}\ln(N))^{1+c} \leq n$ but instead with $\ln(\theta^2 n/\ln(|\mathcal{H}|))$ replaced by $\ln(n)$. A careful examination of their proof reveals the above more general form.

Comparing the best upper and lower bounds, an intriguing gap of $\sqrt{\ln(n)/\ln(e/\mathcal{L}^\theta_\mathbf{S}(f))}$ remains. In particular, for voting classifiers $f$ where a constant fraction of training points have margin less than $\theta$, the gap is a $\sqrt{\ln(n)}$ factor.

**Our Contribution**

In this paper, we finally settle the generalization performance of voting classifiers over finite base hypothesis sets $\mathcal{H}$ by proving an improved generalization upper bound. Our new upper bound matches the lower bound in (3) across the range of the parameters $\theta, \mathcal{L}^\theta_\mathbf{S}(f), n$ and $\delta$ and is as follows

**Theorem 1.1.** *There exists a constant $c > 0$ such that the following holds. Let $\mathcal{D}$ be a distribution over $\mathcal{X} \times \{-1,+1\}$ and let $\mathcal{H} \subseteq \{-1,+1\}^\mathcal{X}$ be a finite base hypothesis set. Then for $n \geq c$ and $0 < \delta < 1$, it holds with probability at least $1 - \delta$ over a sample $\mathbf{S} \sim \mathcal{D}^n$, that for all voting classifiers $f \in \mathcal{C}(\mathcal{H})$ and margins $\theta \in \left(\sqrt{e\ln(|\mathcal{H}|)/n}, 1\right]$ we have:*

$$\mathcal{L}_\mathcal{D}(f) \leq \mathcal{L}^\theta_\mathbf{S}(f)$$
$$+ c\cdot\sqrt{\mathcal{L}^\theta_\mathbf{S}(f)\left(\frac{\ln(e/\mathcal{L}^\theta_\mathbf{S}(f))\ln(|\mathcal{H}|)}{\theta^2 n} + \frac{\ln(e/\delta)}{n}\right)}$$
$$+ c\cdot\left(\frac{\ln(\theta^2 n/\ln(|\mathcal{H}|))\ln(|\mathcal{H}|)}{\theta^2 n} + \frac{\ln(e/\delta)}{n}\right).$$

While it is reasonable to argue that the gap between the previous tightest upper and lower bound is small in magnitude, our result finally settles the generalization performance of

one of the most influential learning techniques, except potentially in the extreme case of $n > 2^{|\mathcal{H}|^c}$ not covered by the lower bound in (3). Additionally, our proof technique elegantly extends a recent novel framework by (Larsen & Schalburg, 2025) for proving generalization of large-margin halfspaces, demonstrating that the framework can be more broadly applied and hopefully paving the way for further tight generalization bounds.

We now conclude the introduction by a brief discussion of other related work. Following that, we present the high-level ideas of our proof, focusing on the significant new contributions in Section 2. After the proof overview, the fully detailed proof is given in Section 3.

**Other Related Work**

Closest to our work is a sequence of results on margin-based generalization bounds for voting classifiers when $\mathcal{H}$ is not finite, but instead has finite VC-dimension $d$. The works (Schapire et al., 1998; Breiman, 1999; Gao & Zhou, 2013) discussed above also present generalization bounds based on the VC-dimension of $\mathcal{H}$. These were all recently improved by (Høgsgaard & Larsen, 2025), who gave the currently tightest generalization bound of

$$\mathcal{L}_{\mathcal{D}}(f) \leq \mathcal{L}_{\mathbf{S}}^{\theta}(f)$$
$$+ c \cdot \sqrt{\mathcal{L}_{\mathbf{S}}^{\theta}(f) \left( \frac{\Gamma(\theta^2 n/d)d}{\theta^2 n} + \frac{\ln(e/\delta)}{n} \right)}$$
$$+ c \cdot \left( \frac{\Gamma(\theta^2 n/d)d}{\theta^2 n} + \frac{\ln(e/\delta)}{n} \right).$$

where $\Gamma(x) = \ln(x) \ln^2(\ln x)$. In terms of lower bounds, one may take the bound from the finite $\mathcal{H}$ case in (3) and replace $\ln(|\mathcal{H}|)$ by $d$. This shows that the latter $\Gamma(\theta^2 n/d)$ term is tight if we replace $\Gamma(x) = \ln(x) \ln^2(\ln x)$ by $\Gamma(x) = \ln x$. For the former $\Gamma(\theta^2 n/d)$ term however, it is conceivable that an improvement to $\ln(e/\mathcal{L}_{\mathbf{S}}^{\theta}(f))$, like our improvement in the finite $\mathcal{H}$ case, is possible. We have not been able to extend our techniques to this setting, but leave it as an interesting direction for future work.

Since large margins imply generalization, much effort has gone into developing algorithms explicitly focusing on maximizing margins. The sequence of works (Breiman, 1999; Grove & Schuurmans, 1998; Bennet et al., 2000; Rätsch & Warmuth, 2002; Rätsch et al., 2005; Grønlund et al., 2019b) for instance focuses on maximizing the smallest margin.

Finally, let us mention that AdaBoost was originally introduced to address a theoretical question on so-called weak-to-strong learning by (Kearns, 1988; Kearns & Valiant, 1994). Informally, a $\gamma$-weak learner, is a learning algorithm that for any distribution $\mathcal{D}$, when given enough samples from $\mathcal{D}$, guarantees to produce a classifier with accuracy at least

$1/2 + \gamma$. If a $\gamma$-weak learner is used to obtain each base classifier used by AdaBoost, then it is possible to show that AdaBoost eventually produces a voting classifier with all margins $\Omega(\gamma)$, see e.g. (Schapire & Freund, 2012) [Theorem 5.8]. The generalization bounds for large-margin voting classifiers above thus apply to AdaBoost when invoked with margins $\theta = \Omega(\gamma)$ and $\mathcal{L}_{\mathbf{S}}^{\theta}(f) = 0$ when AdaBoost is run for enough iterations with a $\gamma$-weak learner. A lower bound by (Høgsgaard et al., 2023) shows that this analysis via large margins is tight for AdaBoost. However for $\mathcal{H}$ with finite VC-dimension, it is possible to develop alternative boosting algorithms with a generalization error that is a $\ln(\gamma^2 n/d)$ factor better than AdaBoost when using a $\gamma$-weak learner, see (Larsen & Ritzert, 2022; Larsen, 2023; Høgsgaard et al., 2024; Høgsgaard & Larsen, 2025).

## 2. Proof Overview

In this section, we give a high level presentation of the main ideas of our proof of Theorem 1.1 and how we manage to improve over previous works. The proof sketch is not meant to be formal, but only to give the intuition and overall structure of the proof. For clarity we ignore all dependencies on the failure probability $\delta$.

Our proof follows the framework of (Schapire et al., 1998) with techniques adapted from (Larsen & Schalburg, 2025). The main idea of the framework is to randomly discretize a voting classifier $f$ in $\mathcal{C}(\mathcal{H})$ to obtain a hypothesis $g \approx f$ in a small finite set of voting classifiers $\mathcal{C}_N$ to be defined below. Since $\mathcal{C}_N$ is small, a union bound shows that all hypotheses in $\mathcal{C}_N$ generalize well, and since $g \approx f$, so does $f$.

While discretization makes it easier to bound the generalization error, the transition from the hypothesis $f$ to the discretized hypothesis $g$ requires us to carefully analyze the difference in accuracy between $f$ and $g$. Our key improvement comes from a much tighter analysis of this cost of discretization compared to the original analysis of (Schapire et al., 1998). Our approach borrows and extends ideas from (Larsen & Schalburg, 2025), who introduced a novel technique using Rademacher complexity to bound the difference in performance between a large-margin halfspace classifier and a random discretization of it.

RANDOM DISCRETIZATION BY SAMPLING

Following the framework of (Schapire et al., 1998), we randomly discretize the voting classifiers $f \in \mathcal{C}(\mathcal{H})$. We define the set of $N$ unweighted averages from $\mathcal{H}$ as $\mathcal{C}_N = \{\frac{1}{N} \sum_{i=1}^{N} h_i \mid h_i \in \mathcal{H}\}$. Our goal is to randomly discretize any $f \in \mathcal{C}(\mathcal{H})$ to a voting classifier $g \in \mathcal{C}_N$. We do this by associating $f$ with a distribution $\mathcal{Q}_f$ over $\mathcal{C}_N$, in such a way that for all $x \in \mathcal{X}$, we have $\mathbb{E}_{\mathbf{g} \sim \mathcal{Q}_f}[\mathbf{g}(x)] = f(x)$. That is, the expectation of the random discretization $g$ equals $f$.

In more detail, let $f \in \mathcal{C}(\mathcal{H})$ be a voting classifier $f = \sum_{h \in \mathcal{H}} a_h h$. Define a distribution $\mathcal{D}_f$ over $\mathcal{H}$ such that the probability of sampling $h$ using $\mathcal{D}_f$ is $a_h$. Specifically, set $\mathbb{P}_{\mathbf{h} \sim \mathcal{D}_f}[\mathbf{h} = h] = a_h$. Then from $\mathcal{D}_f$ we sample an element $g \sim \mathcal{Q}_f$ by sampling $N$ i.i.d. hypotheses $h_1, \ldots, h_N$ and outputting their average $g = \frac{1}{N} \sum_{i=1}^{N} h_i$. Note that any $g$ in the support of $\mathcal{Q}_f$ thus lies in $\mathcal{C}_N$.

### INITIAL ANALYSIS

Starting just like the original framework of (Schapire et al., 1998), we relate the error $\mathcal{L}_{\mathcal{D}}(f)$ of a voting classifier $f \in \mathcal{C}(\mathcal{H})$ with the half-margin error $\mathcal{L}_{\mathcal{D}}^{\theta/2}(g)$ of the discretized classifier $g \sim \mathcal{Q}_f$. Here we observe that for any $g \in \mathcal{C}_N$, it holds that

$$\mathcal{L}_{\mathcal{D}}(f) = \mathcal{L}_{\mathcal{D}}^{\theta/2}(g) + \mathop{\mathbb{P}}_{(\mathbf{x},\mathbf{y}) \sim \mathcal{D}}[\mathbf{y}g(\mathbf{x}) > \theta/2 \wedge \mathbf{y}f(\mathbf{x}) \leq 0]$$
$$- \mathop{\mathbb{P}}_{(\mathbf{x},\mathbf{y}) \sim \mathcal{D}}[\mathbf{y}g(\mathbf{x}) \leq \theta/2 \wedge \mathbf{y}f(\mathbf{x}) > 0]$$

We do the same for the empirical margin error $\mathcal{L}_{\mathbf{S}}^{\theta}(f)$ and the empirical half-margin error $\mathcal{L}_{\mathbf{S}}^{\theta/2}(g)$:

$$\mathcal{L}_{\mathbf{S}}^{\theta}(f) = \mathcal{L}_{\mathbf{S}}^{\theta/2}(g) + \mathop{\mathbb{P}}_{(\mathbf{x},\mathbf{y}) \sim \mathbf{S}}[\mathbf{y}g(\mathbf{x}) > \theta/2 \wedge \mathbf{y}f(\mathbf{x}) \leq \theta]$$
$$- \mathop{\mathbb{P}}_{(\mathbf{x},\mathbf{y}) \sim \mathbf{S}}[\mathbf{y}g(\mathbf{x}) \leq \theta/2 \wedge \mathbf{y}f(\mathbf{x}) > \theta].$$

Combining these equalities, we get

$$\mathcal{L}_{\mathcal{D}}(f) - \mathcal{L}_{\mathbf{S}}^{\theta}(f) = \mathcal{L}_{\mathcal{D}}^{\theta/2}(g) - \mathcal{L}_{\mathbf{S}}^{\theta/2}(g) \qquad (4)$$
$$+ \mathop{\mathbb{P}}_{(\mathbf{x},\mathbf{y}) \sim \mathcal{D}}[\mathbf{y}g(\mathbf{x}) > \theta/2 \wedge \mathbf{y}f(\mathbf{x}) \leq 0]$$
$$- \mathop{\mathbb{P}}_{(\mathbf{x},\mathbf{y}) \sim \mathbf{S}}[\mathbf{y}g(\mathbf{x}) > \theta/2 \wedge \mathbf{y}f(\mathbf{x}) \leq \theta]$$
$$+ \mathop{\mathbb{P}}_{(\mathbf{x},\mathbf{y}) \sim \mathbf{S}}[\mathbf{y}g(\mathbf{x}) \leq \theta/2 \wedge \mathbf{y}f(\mathbf{x}) > \theta]$$
$$- \mathop{\mathbb{P}}_{(\mathbf{x},\mathbf{y}) \sim \mathcal{D}}[\mathbf{y}g(\mathbf{x}) \leq \theta/2 \wedge \mathbf{y}f(\mathbf{x}) > 0].$$

### PROOF OF (SCHAPIRE ET AL., 1998)

In the original proof of (Schapire et al., 1998), they simply drop the two terms $\mathbb{P}_{(\mathbf{x},\mathbf{y}) \sim \mathcal{D}}[\mathbf{y}g(\mathbf{x}) \leq \theta/2 \wedge \mathbf{y}f(\mathbf{x}) > 0]$ and $\mathbb{P}_{(\mathbf{x},\mathbf{y}) \sim \mathbf{S}}[\mathbf{y}g(\mathbf{x}) > \theta/2 \wedge \mathbf{y}f(\mathbf{x}) \leq \theta]$. Taking expectation over $g \sim \mathcal{Q}_f$ on both sides of the inequality then gives

$$\mathcal{L}_{\mathcal{D}}(f) - \mathcal{L}_{\mathbf{S}}^{\theta}(f)$$
$$\leq \mathop{\mathbb{E}}_{\mathbf{g}}[\mathcal{L}_{\mathcal{D}}^{\theta/2}(\mathbf{g}) - \mathcal{L}_{\mathbf{S}}^{\theta/2}(\mathbf{g})]$$
$$+ \mathop{\mathbb{E}}_{\mathbf{g}}[\mathop{\mathbb{P}}_{(\mathbf{x},\mathbf{y}) \sim \mathcal{D}}[\mathbf{y}g(\mathbf{x}) > \theta/2 \wedge \mathbf{y}f(\mathbf{x}) \leq 0]]$$
$$+ \mathop{\mathbb{E}}_{\mathbf{g}}[\mathop{\mathbb{P}}_{(\mathbf{x},\mathbf{y}) \sim \mathbf{S}}[\mathbf{y}g(\mathbf{x}) \leq \theta/2 \wedge \mathbf{y}f(\mathbf{x}) > \theta]].$$

Here we used that the left hand side is independent of $g$ and thus the expectation may be dropped. Using that $\mathcal{C}_N$

is small in combination with Hoeffding's inequality and a union bound gives

$$\mathop{\mathbb{E}}_{\mathbf{g}}[\mathcal{L}_{\mathcal{D}}^{\theta/2}(\mathbf{g}) - \mathcal{L}_{\mathbf{S}}^{\theta/2}(\mathbf{g})] \leq \sup_{g \in \mathcal{C}_N}[\mathcal{L}_{\mathcal{D}}^{\theta/2}(g) - \mathcal{L}_{\mathbf{S}}^{\theta/2}(g)]$$
$$\leq c\sqrt{\ln(|\mathcal{C}_N|)/n}$$
$$= c\sqrt{N \ln(|\mathcal{H}|)/n}. \qquad (5)$$

The remaining two terms are handled symmetrically, so let us focus on $\mathbb{E}_{\mathbf{g}}[\mathbb{P}_{(\mathbf{x},\mathbf{y}) \sim \mathbf{S}}[\mathbf{y}g(\mathbf{x}) \leq \theta/2 \wedge \mathbf{y}f(\mathbf{x}) > \theta]]$. Swapping the order of expectation and probability gives

$$\mathop{\mathbb{E}}_{\mathbf{g}}[\mathop{\mathbb{P}}_{(\mathbf{x},\mathbf{y}) \sim \mathbf{S}}[\mathbf{y}g(\mathbf{x}) \leq \theta/2 \wedge \mathbf{y}f(\mathbf{x}) > \theta]]$$
$$= \mathop{\mathbb{E}}_{(\mathbf{x},\mathbf{y}) \sim \mathbf{S}}[\mathop{\mathbb{P}}_{\mathbf{g} \sim \mathcal{Q}_f}[\mathbf{y}g(\mathbf{x}) \leq \theta/2 \wedge \mathbf{y}f(\mathbf{x}) > \theta]]$$
$$\leq \mathop{\mathbb{E}}_{(\mathbf{x},\mathbf{y}) \sim \mathbf{S}}[\mathop{\mathbb{P}}_{\mathbf{g} \sim \mathcal{Q}_f}[|\mathbf{g}(\mathbf{x}) - f(\mathbf{x})| > \theta/2]].$$

Now observe that for any $x$, the prediction $\mathbf{g}(x)$ when $\mathbf{g} \sim \mathcal{Q}_f$ is obtained by averaging $N$ i.i.d. predictions in $\{-1, +1\}$, each with expectation $f(x)$. By Hoeffding's inequality, we thus have

$$\mathop{\mathbb{E}}_{\mathbf{g}}[\mathop{\mathbb{P}}_{(\mathbf{x},\mathbf{y}) \sim \mathbf{S}}[\mathbf{y}g(\mathbf{x}) \leq \theta/2 \wedge \mathbf{y}f(\mathbf{x}) > \theta]]$$
$$\leq 2\exp(-\theta^2 N/8).$$

Combining it all, (Schapire et al., 1998) conclude

$$\mathcal{L}_{\mathcal{D}}(f) - \mathcal{L}_{\mathbf{S}}^{\theta}(f) \leq c\left(\sqrt{N \ln(|\mathcal{H}|)/n} + \exp(-\theta^2 N/8)\right).$$

Choosing $N \approx \theta^{-2} \ln n$ finally gives $\mathcal{L}_{\mathcal{D}}(f) - \mathcal{L}_{\mathbf{S}}^{\theta}(f) \leq c\sqrt{\ln(|\mathcal{H}|)\ln(n)/(\theta^2 n)}$.

### OUR KEY IMPROVEMENTS

Our new proof departs from (Schapire et al., 1998) after the equality (4). Where they simply drop the two negative terms, we carefully exploit these to get our improvement. Repeating the expectation over $\mathbf{g} \sim \mathcal{Q}_f$, we get

$$\mathcal{L}_{\mathcal{D}}(f) - \mathcal{L}_{\mathbf{S}}^{\theta}(f) \qquad (6)$$
$$\leq \mathop{\mathbb{E}}_{\mathbf{g}}[\mathcal{L}_{\mathcal{D}}^{\theta/2}(\mathbf{g}) - \mathcal{L}_{\mathbf{S}}^{\theta/2}(\mathbf{g})]$$
$$+ \mathop{\mathbb{E}}_{\mathbf{g}}[\mathop{\mathbb{P}}_{(\mathbf{x},\mathbf{y}) \sim \mathcal{D}}[\mathbf{y}g(\mathbf{x}) > \theta/2 \wedge \mathbf{y}f(\mathbf{x}) \leq 0]]$$
$$- \mathop{\mathbb{E}}_{\mathbf{g}}[\mathop{\mathbb{P}}_{(\mathbf{x},\mathbf{y}) \sim \mathbf{S}}[\mathbf{y}g(\mathbf{x}) > \theta/2 \wedge \mathbf{y}f(\mathbf{x}) \leq \theta]]$$
$$+ \mathop{\mathbb{E}}_{\mathbf{g}}[\mathop{\mathbb{P}}_{(\mathbf{x},\mathbf{y}) \sim \mathbf{S}}[\mathbf{y}g(\mathbf{x}) \leq \theta/2 \wedge \mathbf{y}f(\mathbf{x}) > \theta]]$$
$$- \mathop{\mathbb{E}}_{\mathbf{g}}[\mathop{\mathbb{P}}_{(\mathbf{x},\mathbf{y}) \sim \mathcal{D}}[\mathbf{y}g(\mathbf{x}) \leq \theta/2 \wedge \mathbf{y}f(\mathbf{x}) > 0]].$$

If we carefully group the different $f \in \mathcal{C}(\mathcal{H})$ based on $\mathcal{L}_{\mathcal{D}}^{\theta}(f)$ and use Chernoff/Bernstein instead of Hoeffding's inequality, we may again union bound over $\mathcal{C}_N$ to conclude

$$\mathop{\mathbb{E}}_{\mathbf{g}}[\mathcal{L}_{\mathcal{D}}^{\theta/2}(\mathbf{g}) - \mathcal{L}_{\mathbf{S}}^{\theta/2}(\mathbf{g})] \leq c\sqrt{\mathcal{L}_{\mathbf{S}}^{\theta}(f) N \ln(|\mathcal{H}|)/n}.$$

The grouping and use of Chernoff/Bernstein thus buys us the factor $\mathcal{L}_{\mathbf{S}}^{\theta}(f)$ compared to (5). This is as such not particularly novel and could also have been carried out in the original proof of (Schapire et al., 1998).

The four remaining lines in (6) are where we make novel use of the recent ideas of (Larsen & Schalburg, 2025). Each pair of lines is handled symmetrically, so we focus on the first one in this overview. We see that

$$
\mathop{\mathbb{E}}_{\mathbf{g}\sim\mathcal{Q}_f}\left[\mathop{\mathbb{P}}_{(\mathbf{x},\mathbf{y})\sim\mathcal{D}}[\mathbf{y}\mathbf{g}(\mathbf{x}) > \theta/2 \wedge \mathbf{y}f(\mathbf{x}) \leq 0]\right]
$$
$$
- \mathop{\mathbb{E}}_{\mathbf{g}\sim\mathcal{Q}_f}\left[\mathop{\mathbb{P}}_{(\mathbf{x},\mathbf{y})\sim\mathbf{S}}[\mathbf{y}\mathbf{g}(\mathbf{x}) > \theta/2 \wedge \mathbf{y}f(\mathbf{x}) \leq \theta]\right]
$$
$$
= \mathop{\mathbb{E}}_{(\mathbf{x},\mathbf{y})\sim\mathcal{D}}\left[\mathop{\mathbb{P}}_{\mathbf{g}\sim\mathcal{Q}_f}[\mathbf{y}\mathbf{g}(\mathbf{x}) > \theta/2 \wedge \mathbf{y}f(\mathbf{x}) \leq 0]\right]
$$
$$
- \mathop{\mathbb{E}}_{(\mathbf{x},\mathbf{y})\sim\mathbf{S}}\left[\mathop{\mathbb{P}}_{\mathbf{g}\sim\mathcal{Q}_f}[\mathbf{y}\mathbf{g}(\mathbf{x}) > \theta/2 \wedge \mathbf{y}f(\mathbf{x}) \leq \theta]\right]. \quad (7)
$$

Notice that (7) is a difference in expectation over the data distribution $\mathcal{D}$ and the training sample $\mathbf{S} \sim \mathcal{D}^n$. The familiar reader may recall that Rademacher complexity provides a powerful tool for bounding such differences when considering the expectation of the same function $\phi(\mathbf{x},\mathbf{y})$ over $(\mathbf{x},\mathbf{y}) \sim \mathcal{D}$ and $(\mathbf{x},\mathbf{y}) \sim \mathbf{S}$. Unfortunately, the two functions inside the expectations in (7) are not identical. A critical observation here is that the distribution of the margin of $\mathbf{g} \sim \mathcal{Q}_f$ on a point $(x,y)$ depends only on the margin of $f$ on the same point. In particular, smaller margins for $f$ reduces the probability that the margin of $\mathbf{g}$ is large and thus

$$
\mathop{\mathbb{P}}_{\mathbf{g}\sim\mathcal{Q}_f}[\mathbf{y}\mathbf{g}(\mathbf{x}) > \theta/2 \wedge \mathbf{y}f(\mathbf{x}) \leq \theta]
$$
$$
\geq \mathop{\mathbb{P}}_{\mathbf{g}\sim\mathcal{Q}_f}[\mathbf{y}\mathbf{g}(\mathbf{x}) > \theta/2 \wedge \mathbf{y}f(\mathbf{x}) \leq 0].
$$

Again exploiting that the distribution of $y\mathbf{g}(x)$ with $\mathbf{g} \sim \mathcal{Q}_f$ is determined solely from $yf(x)$, we can now define the function $\phi : \mathbb{R} \to \mathbb{R}$ so that $\phi(yf(x)) = \mathbb{1}_{\{yf(x)\leq 0\}}\mathbb{P}_{\mathbf{g}\sim\mathcal{Q}_f}[y\mathbf{g}(x) > \theta/2]$. We then have that

$$
(7) \leq \mathop{\mathbb{E}}_{(\mathbf{x},\mathbf{y})\sim\mathcal{D}}[\phi(\mathbf{y}f(\mathbf{x}))] - \mathop{\mathbb{E}}_{(\mathbf{x},\mathbf{y})\sim\mathbf{S}}[\phi(\mathbf{y}f(\mathbf{x}))].
$$

We are now in a position where we can use Rademacher complexity. In particular, the seminal Contraction inequality of (Ledoux & Talagrand, 1991) may be applied, allowing us to remove $\phi$ at the cost of multiplying with its Lipschitz constant $L$. This of course requires $\phi$ to be a Lipschitz continuous function with a bounded Lipschitz constant $L$. Unfortunately the $\phi$ defined above is discontinuous due to the multiplication with the indicator function. After some careful massaging of $\phi$, we end up being able to bound its Lipschitz constant by $L \leq c\theta^{-1}\exp(-N\theta^2/c)$. Using the Contraction inequality this gives

$$
(7) \leq c\theta^{-1}\exp(-N\theta^2/c)
$$
$$
\cdot \sup_{f\in\mathcal{C}(\mathcal{H})}[\mathop{\mathbb{E}}_{(\mathbf{x},\mathbf{y})\sim\mathcal{D}}[\mathbf{y}f(\mathbf{x})] - \mathop{\mathbb{E}}_{(\mathbf{x},\mathbf{y})\sim\mathbf{S}}[\mathbf{y}f(\mathbf{x})]].
$$

Using that every $f \in \mathcal{C}(\mathcal{H})$ is a convex combination of hypotheses $h \in \mathcal{H}$, we get that the sup is bounded by a $\sup_{h\in\mathcal{H}}$, i.e.

$$
(7) \leq c\theta^{-1}\exp(-N\theta^2/c)
$$
$$
\cdot \sup_{h\in\mathcal{H}}[\mathop{\mathbb{E}}_{(\mathbf{x},\mathbf{y})\sim\mathcal{D}}[\mathbf{y}h(\mathbf{x})] - \mathop{\mathbb{E}}_{(\mathbf{x},\mathbf{y})\sim\mathbf{S}}[\mathbf{y}h(\mathbf{x})]].
$$

Using the finite class lemma of (Massart, 2000) finally gives

$$
(7) \leq c\theta^{-1}\exp(-N\theta^2/c) \cdot \sqrt{\frac{\ln(|\mathcal{H}|)}{n}}
$$
$$
= c\exp(-N\theta^2/c) \cdot \sqrt{\frac{\ln(|\mathcal{H}|)}{\theta^2 n}}.
$$

If we compare this to the proof of (Schapire et al., 1998), we have thus reduced the contribution of the two last lines of (6) by a factor $c\sqrt{\ln(|\mathcal{H}|)/n}$. Combining it all, we have shown

$$
\mathcal{L}_{\mathcal{D}}(f) - \mathcal{L}_{\mathbf{S}}^{\theta}(f) \leq c \cdot \sqrt{\frac{\mathcal{L}_{\mathbf{S}}^{\theta}(f)N\ln(|\mathcal{H}|)}{n}}
$$
$$
+ c \cdot \exp(-N\theta^2/c) \cdot \sqrt{\frac{\ln(|\mathcal{H}|)}{\theta^2 n}}.
$$

Picking $N \approx \theta^{-2}\ln(e/\mathcal{L}_{\mathbf{S}}^{\theta}(f))$ balances the terms and finally completes the proof.

## 3. Main Proof

We now start on the proof of Theorem 1.1, before doing the main part of work, we do a series of reductions which allows us to focus on establishing the theorem for margins $\theta$ and empirical margin-errors $\mathcal{L}_{\mathbf{S}}^{\theta}(f)$ in small ranges at a time. We start by establishing these reductions in Subsection 3.1, and continue with the main body of the proof in Subsection 3.2.

### 3.1. Preliminaries

STATEMENT REDUCTION

Firstly, to ease later analysis, we want to ensure margins in $[-c_\theta, c_\theta]$ for a constant $c_\theta < 1$. which we do by scaling all voting classifiers by $c_\theta$ and adding the all-one and all-negative-one with appropriate constants afterwards, the argument can be found in full detail in Appendix D.1.

From now on, let $\mathcal{D}$ be an arbitrary distribution over $\mathcal{X} \times \{-1,+1\}$, we want to prove that there is a constant $c > 0$, such that with probability at least $1 - \delta$ over the training sample $\mathbf{S} \sim \mathcal{D}^n$, it holds for all margins $\theta \in \left(\sqrt{e\ln(|\mathcal{H}|)/n}, 1\right]$ and all $f \in \mathcal{C}(\mathcal{H})$ that:

$$\mathcal{L}_{\mathcal{D}}(f) \leq \mathcal{L}_{\mathbf{S}}^{\theta}(f) \tag{8}$$

$$+ c \cdot \sqrt{\mathcal{L}_{\mathbf{S}}^{\theta}(f)\left(\frac{\ln(e/\mathcal{L}_{\mathbf{S}}^{\theta}(f))\ln(|\mathcal{H}|)}{\theta^2 n} + \frac{\ln(e/\delta)}{n}\right)}$$

$$+ c \cdot \left(\frac{\ln(\theta^2 n/\ln(|\mathcal{H}|))\ln(|\mathcal{H}|)}{\theta^2 n} + \frac{\ln(e/\delta)}{n}\right)$$

Theorem 1.1 follows as a corollary.

SPLITTING INTO SMALLER TASKS

We break the task of proving (8) into smaller tasks, which then gives the full statement after union bounding over the sub-tasks.

Firstly we partition the sets of possible margins and margin losses into collections of manageable subsets, we define:

$$\Theta_i = \left(e2^{i-1}\sqrt{\frac{\ln|\mathcal{H}|}{n}}, e2^i\sqrt{\frac{\ln|\mathcal{H}|}{n}}\right],$$

$$L_0 = [0, n^{-1}] \text{ and } L_j = \left(2^{i-1}n^{-1}, 2^i n^{-1}\right]$$

Then the collections $\{\Theta_i\}_{i=1}^{\log_2((e/c_\theta)\sqrt{n/\ln|\mathcal{H}|})}, \{L_j\}_{j=0}^{\log_2(n)}$ partition the sets of possible margins $\theta \in \left(\sqrt{e\ln(|\mathcal{H}|)/n}, 1\right]$ and true margin losses $\mathcal{L}_{\mathcal{D}}^{\theta}(f) \in [0, 1]$.

For each pair of parameter sets $(\Theta_i, L_j)$, where $\Theta_i = (\theta_i, \theta_{i+1}]$, we define a corresponding set of voting classifiers:

$$\mathcal{C}_{\mathcal{H}}(\Theta_i, L_j) = \left\{f \in \mathcal{C}(\mathcal{H}) \ \bigg| \ \mathcal{L}_{\mathcal{D}}^{(3/4)\theta_i}(f) \in L_j\right\}$$

We now want to prove a version of (8), but tailored to a pair of parameter sets $(\Theta_i, L_j)$.

**Lemma 3.1.** *There exists a constant $c > 0$ such that the following holds. For any $0 < \delta < 1$, and $(\Theta_i, L_j) = ((\theta_i, \theta_{i+1}], (l_j, l_{j+1}))$ it holds with probability at least $1 - \delta$ over a the sample $\mathbf{S} \sim \mathcal{D}^n$, that:*

$$\sup_{\substack{f \in \mathcal{C}_{\mathcal{H}}(\Theta_i, L_j) \\ \theta \in \Theta_i}} |\mathcal{L}_{\mathcal{D}}(f) - \mathcal{L}_{\mathbf{S}}^{\theta}(f)| \tag{9}$$

$$\leq c \cdot \sqrt{l_{j+1}\left(\frac{\ln(e/l_{j+1})\ln(|\mathcal{H}|)}{\theta_{i+1}^2 n} + \frac{\ln(e/\delta)}{n}\right)}$$

$$+ c \cdot \left(\frac{\ln(e/l_{j+1})\ln(|\mathcal{H}|)}{\theta_{i+1}^2 n} + \frac{\ln(e/\delta)}{n}\right).$$

For this to be enough to prove (8), we need to relate $l_{j+1}$ to $\mathcal{L}_{\mathbf{S}}^{\theta}(f)$, which we do using the following lemma

**Lemma 3.2.** *There is a constant $c > 0$, such that for any $0 < \delta < 1$ and any $\Theta_i = (\theta_i, \theta_{i+1}]$, it holds with probability*

at least $1 - \delta$ over the sample $\mathbf{S} \sim \mathcal{D}^n$

$$\forall f \in \mathcal{C}(\mathcal{H}): \tag{10}$$

$$\frac{\mathcal{L}_{\mathcal{D}}^{(3/4)\theta_i}(f)}{2} - \mathcal{L}_{\mathbf{S}}^{\theta_i}(f)$$

$$\leq c\left(\frac{\ln(\theta_{i+1}^2 n/\ln(|\mathcal{H}|))\ln(|\mathcal{H}|)}{\theta_{i+1}^2 n} + \frac{\ln(e/\delta)}{n}\right)$$

*When $N \geq 64 \cdot \theta_{i+1}^{-2}$.*

To use this lemma, together with Lemma 3.1, we need them to hold simultaneously for all pairs $(\Theta_i, L_j)$, after which we can use them together and obtain (8).

**Claim 3.3.** *For any $0 < \delta < 1$, the following holds:*

1. *If the sample $\mathbf{S} \sim \mathcal{D}^n$ satisfies (9) and (10) simultaneously for all $(\Theta_i, L_j)$ and $\Theta_i$, with slightly different constants, then for that sample, (8) holds for all margins $\theta \in \left(\sqrt{e\ln(|\mathcal{H}|)/n}, 1\right]$ and all $f \in \mathcal{C}(\mathcal{H})$.*

2. *With probability at least $1 - \delta$ over the sample $\mathbf{S} \sim \mathcal{D}^n$, the above event happens*

The proof of Claim 3.3 can be found in Appendix D.1. To prove the first part, Lemma 3.2 is used to relate $l_{j+1}$ and $\mathcal{L}_{\mathbf{S}}^{\theta}(f)$ in Lemma 3.1. To prove the second part, a simple union bound is enough.

The two parts of Claim 3.3 together finishes the proof. It remains to prove Lemmas 3.2 and 3.1. The proof of Lemma 3.2 can be found in Appendix C, as it is standard. We proceed with the final part of the proof, the proof of Lemma 3.1.

### 3.2. Proof

Now we focus on the proof of Lemma 3.1.

So let $0 < \delta < 1$ and fix a pair $(\Theta_i, L_j)$. We follow the process described in Section 2, i.e we want to discretize a voting classifier $f \in \mathcal{C}(\mathcal{H})$ to a unweighted average $g$ in $\mathcal{C}_N = \{\frac{1}{N}\sum_{i=1}^N h_i \mid h_i \in \mathcal{H}\}$.

For $f \in \mathcal{C}(\mathcal{H})$, given by $f = \sum_{h \in \mathcal{H}} a_h h$, define distribution $\mathcal{D}_f$ over $\mathcal{H}$ by setting the probability of sampling $h$ to $a_h$. Let $\mathcal{Q}_f$ be the distribution of a $\mathbf{g} \in \mathcal{C}_N$ obtained by sampling $N$ i.i.d. $\mathbf{h}_1, \ldots, \mathbf{h}_N \in \mathcal{H}$ using $\mathcal{D}_f$ and taking their average $\mathbf{g} = \frac{1}{N}\sum_{i=1}^N \mathbf{h}_i$.

Define $\mathcal{L}_{\mathcal{D}}^{\theta}(f) = \mathbb{P}_{(\mathbf{x}, \mathbf{y}) \sim \mathcal{D}}[\mathbf{y}f(\mathbf{x}) \leq \theta]$ to be the probability that the margin of a point $(\mathbf{x}, \mathbf{y}) \sim \mathcal{D}$ is less than $\theta$. And write $\mathcal{L}_{\mathcal{D}}(f)$ for $\mathcal{L}_{\mathcal{D}}^0(f)$.

We want to relate the true loss $\mathcal{L}_{\mathcal{D}}(f)$ of $f$, to the margin loss $\mathcal{L}_{\mathcal{D}}^{\theta/2}(\mathbf{g})$ of a $\mathbf{g} \sim \mathcal{Q}_f$. Firstly for every $\theta \in \Theta_i, g \in \mathcal{C}_N$

and training sample $S$, it holds that:

$$\mathcal{L}_{\mathcal{D}}(f) = \mathcal{L}_{\mathcal{D}}^{\theta_i/2}(g)$$
$$+ \mathop{\mathbb{P}}_{(\mathbf{x},\mathbf{y})\sim\mathcal{D}}[\mathbf{y}g(\mathbf{x}) > \theta_i/2 \wedge \mathbf{y}f(\mathbf{x}) \le 0]$$
$$- \mathop{\mathbb{P}}_{(\mathbf{x},\mathbf{y})\sim\mathcal{D}}[\mathbf{y}g(\mathbf{x}) \le \theta_i/2 \wedge \mathbf{y}f(\mathbf{x}) > 0],$$

and

$$\mathcal{L}_{S}^{\theta}(f) = \mathcal{L}_{S}^{\theta_i/2}(g)$$
$$+ \mathop{\mathbb{P}}_{(\mathbf{x},\mathbf{y})\sim S}[\mathbf{y}g(\mathbf{x}) > \theta_i/2 \wedge \mathbf{y}f(\mathbf{x}) \le \theta]$$
$$- \mathop{\mathbb{P}}_{(\mathbf{x},\mathbf{y})\sim S}[\mathbf{y}g(\mathbf{x}) \le \theta_i/2 \wedge \mathbf{y}f(\mathbf{x}) > \theta].$$

This allows us to split the difference $\mathcal{L}_{\mathcal{D}}(f) - \mathcal{L}_{S}^{\theta}(f)$ into the sum of 3 differences:

$$\mathcal{L}_{\mathcal{D}}(f) - \mathcal{L}_{S}^{\theta}(f) = \mathcal{L}_{\mathcal{D}}^{\theta_i/2}(g) - \mathcal{L}_{S}^{\theta_i/2}(g)$$
$$+ \mathop{\mathbb{P}}_{(\mathbf{x},\mathbf{y})\sim\mathcal{D}}[\mathbf{y}g(\mathbf{x}) > \theta_i/2 \wedge \mathbf{y}f(\mathbf{x}) \le 0]$$
$$- \mathop{\mathbb{P}}_{(\mathbf{x},\mathbf{y})\sim S}[\mathbf{y}g(\mathbf{x}) > \theta_i/2 \wedge \mathbf{y}f(\mathbf{x}) \le \theta]$$
$$+ \mathop{\mathbb{P}}_{(\mathbf{x},\mathbf{y})\sim S}[\mathbf{y}g(\mathbf{x}) \le \theta_i/2 \wedge \mathbf{y}f(\mathbf{x}) > \theta]$$
$$- \mathop{\mathbb{P}}_{(\mathbf{x},\mathbf{y})\sim\mathcal{D}}[\mathbf{y}g(\mathbf{x}) \le \theta_i/2 \wedge \mathbf{y}f(\mathbf{x}) > 0]$$

Taking expectation over $\mathbf{g} \sim \mathcal{Q}_f$ and then supremum over $f \in \mathcal{C}_{\mathcal{H}}(\Theta_i, L_j)$, gives:

$$\sup_{f\in\mathcal{C}_{\mathcal{H}}(\Theta_i,L_j)} \left(\mathcal{L}_{\mathcal{D}}(f) - \mathcal{L}_{S}^{\theta}(f)\right)$$

$$= \sup_{f\in\mathcal{C}_{\mathcal{H}}(\Theta_i,L_j)} \left( \mathop{\mathbb{E}}_{\mathbf{g}\sim\mathcal{Q}_f}\left[\mathcal{L}_{\mathcal{D}}^{\theta_i/2}(\mathbf{g}) - \mathcal{L}_{\mathbf{S}}^{\theta_i/2}(\mathbf{g})\right] \right.$$

$$+ \mathop{\mathbb{E}}_{\mathbf{g}\sim\mathcal{Q}_f}\left[ \mathop{\mathbb{P}}_{(\mathbf{x},\mathbf{y})\sim\mathcal{D}}[\mathbf{y}g(\mathbf{x}) > \theta_i/2 \wedge \mathbf{y}f(\mathbf{x}) \le 0] \right.$$

$$\left. - \mathop{\mathbb{P}}_{(\mathbf{x},\mathbf{y})\sim S}[\mathbf{y}g(\mathbf{x}) > \theta_i/2 \wedge \mathbf{y}f(\mathbf{x}) \le \theta]\right] \tag{11}$$

$$+ \mathop{\mathbb{E}}_{\mathbf{g}\sim\mathcal{Q}_f}\left[ \mathop{\mathbb{P}}_{(\mathbf{x},\mathbf{y})\sim S}[\mathbf{y}g(\mathbf{x}) \le \theta_i/2 \wedge \mathbf{y}f(\mathbf{x}) > \theta]\right.$$

$$\left.\left. - \mathop{\mathbb{P}}_{(\mathbf{x},\mathbf{y})\sim\mathcal{D}}[\mathbf{y}g(\mathbf{x}) \le \theta_i/2 \wedge \mathbf{y}f(\mathbf{x}) > 0]\right] \right). \tag{12}$$

To bound the last two terms, we define two continuous functions: $\phi, \rho : [-c_\theta, c_\theta] \to \mathbb{R}_{\ge 0}$

$$\phi(\lambda) =$$
$$\begin{cases} \mathbb{P}[y\mathbf{g}(x) > \theta_i/2 \mid yf(x) = \lambda] & \lambda \in (-c_\theta, 0] \\ \frac{\theta_i - \lambda}{\theta_i}\,\mathbb{P}[y\mathbf{g}(x) > \theta_i/2 \mid yf(x) = 0] & \lambda \in (0, \theta_i] \\ 0 & \lambda \in (-\theta_i, c_\theta] \end{cases}$$

$$\rho(\lambda) =$$
$$\begin{cases} 0 & \lambda \in (c_\theta, 0] \\ \frac{\lambda}{\theta_i}\,\mathbb{P}[y\mathbf{g}(x) \le \theta_i/2 \mid yf(x) = \theta_i] & \lambda \in (0, \theta_i] \\ \mathbb{P}[y\mathbf{g}(x) \le \theta_i/2 \mid yf(x) = \lambda] & \lambda \in (\theta_i, c_\theta] \end{cases}$$

Note that we abuse notation slightly in the above by writing conditioned on $yf(x) = \lambda$. Here we use that the distribution of $yg(x)$ for $\mathbf{g} \sim \mathcal{Q}_f$ is determined solely from the value $yf(x)$. Thus when we write conditioned on $yf(x) = \lambda$ we implicitly mean for any $(x, y)$ and $f$ with $yf(x) = \lambda$.

Now we can replace (11) and (12) with a difference in expectations of $\phi$ and $\rho$, because they upper bound the terms

**Lemma 3.4.**

$$\mathop{\mathbb{E}}_{\mathbf{g}\sim\mathcal{Q}_f}\left[ \mathop{\mathbb{P}}_{(\mathbf{x},\mathbf{y})\sim\mathcal{D}}[\mathbf{y}g(\mathbf{x}) > \theta_i/2 \wedge \mathbf{y}f(\mathbf{x}) \le 0]\right]$$
$$\le \mathop{\mathbb{E}}_{(\mathbf{x},\mathbf{y})\sim\mathcal{D}}[\phi(\mathbf{y}f(\mathbf{x}))]$$

$$\mathop{\mathbb{E}}_{\mathbf{g}\sim\mathcal{Q}_f}\left[ \mathop{\mathbb{P}}_{(\mathbf{x},\mathbf{y})\sim S}[\mathbf{y}g(\mathbf{x}) > \theta_i/2 \wedge \mathbf{y}f(\mathbf{x}) \le \theta]\right]$$
$$\ge \mathop{\mathbb{E}}_{(\mathbf{x},\mathbf{y})\sim S}[\phi(\mathbf{y}f(\mathbf{x}))]$$

$$\mathop{\mathbb{E}}_{\mathbf{g}\sim\mathcal{Q}_f}\left[ \mathop{\mathbb{P}}_{(\mathbf{x},\mathbf{y})\sim S}[\mathbf{y}g(\mathbf{x}) \le \theta_i/2 \wedge \mathbf{y}f(\mathbf{x}) > \theta]\right]$$
$$\le \mathop{\mathbb{E}}_{(\mathbf{x},\mathbf{y})\sim S}[\rho(\mathbf{y}f(\mathbf{x}))]$$

$$\mathop{\mathbb{E}}_{\mathbf{g}\sim\mathcal{Q}_f}\left[ \mathop{\mathbb{P}}_{(\mathbf{x},\mathbf{y})\sim\mathcal{D}}[\mathbf{y}g(\mathbf{x}) \le \theta_i/2 \wedge \mathbf{y}f(\mathbf{x}) > 0]\right]$$
$$\ge \mathop{\mathbb{E}}_{(\mathbf{x},\mathbf{y})\sim\mathcal{D}}[\rho(\mathbf{y}f(\mathbf{x}))]$$

The statement follows from the definitions of $\phi$ and $\rho$ as well as monotonicity of the conditional properties in their definitions, the proof can be found in Section D: Deferred proofs.

The statement, together with linearity of expectation, subadditivity of supremum and the triangle inequality, gives us an upper bound

$$\sup_{f\in\mathcal{C}_{\mathcal{H}}(\Theta_i,L_j)} \left(\mathcal{L}_{\mathcal{D}}(f) - \mathcal{L}_{S}^{\theta}(f)\right)$$

$$\le \sup_{f\in\mathcal{C}_{\mathcal{H}}(\Theta_i,L_j)} \left| \mathop{\mathbb{E}}_{\mathbf{g}\sim\mathcal{Q}_f}\left[\mathcal{L}_{\mathcal{D}}^{\theta_i/2}(\mathbf{g}) - \mathcal{L}_{S}^{\theta_i/2}(\mathbf{g})\right] \right| \tag{13}$$

$$+ \sup_{f\in\mathcal{C}_{\mathcal{H}}(\Theta_i,L_j)} \left| \mathop{\mathbb{E}}_{(\mathbf{x},\mathbf{y})\sim\mathcal{D}}[\phi(\mathbf{y}f(\mathbf{x}))] - \mathop{\mathbb{E}}_{(\mathbf{x},\mathbf{y})\sim S}[\phi(\mathbf{y}f(\mathbf{x}))] \right| \tag{14}$$

$$+ \sup_{f\in\mathcal{C}_{\mathcal{H}}(\Theta_i,L_j)} \left| \mathop{\mathbb{E}}_{(\mathbf{x},\mathbf{y})\sim\mathcal{D}}[\rho(\mathbf{y}f(\mathbf{x}))] - \mathop{\mathbb{E}}_{(\mathbf{x},\mathbf{y})\sim S}[\rho(\mathbf{y}f(\mathbf{x}))] \right| \tag{15}$$

The rest of the proof comes down to handle these differences, we will handle the first term and the last two separately. In Section B we use standard techniques to bound (13), as described by the lemma:

**Lemma 3.5.** *There exists $c > 0$ such that with probability greater than $1 - \delta$ over the sample $\mathbf{S} \sim \mathcal{D}^n$ it holds that:*

$$
\sup_{f \in \mathcal{C}_{\mathcal{H}}(\Theta_i, L_j)} \left( \left| \mathbb{E}_{\mathbf{g} \sim \mathcal{Q}_f}[\mathcal{L}_{\mathcal{D}}^{\theta_i/2}(\mathbf{g}) - \mathcal{L}_{\mathbf{S}}^{\theta_i/2}(\mathbf{g})] \right| \right)
$$
$$
\leq c \left( \sqrt{\frac{(l_{j+1} + \exp\left(-N\theta_{i+1}^2/c\right))\ln(|\mathcal{H}|^N/\delta)}{n}} \right.
$$
$$
\left. + \frac{\ln(|\mathcal{H}|^N/\delta)}{n} \right)
$$

*when $N \geq 32 \cdot \theta_{i+1}^{-2}$.*

In Section A we use Rademacher complexity theory together with a bound on the Lipschitz constants of $\phi, \rho$ to bound (14) and (15), as described by the lemma

**Lemma 3.6.** *There exists a constant $c > 0$ such that when $N \geq 32 \cdot \theta_{i+1}^{-2}$ it holds with probability at least $1 - \delta$ over the sample $\mathbf{S} \sim \mathcal{D}^n$ that:*

$$
\max\{(14), (15)\} \leq c \exp\left(-\frac{\theta_{i+1}^2 N}{c}\right) \cdot
$$
$$
\left( \sqrt{\frac{(\theta_{i+1}^2 N^2 + \theta_{i+1}^{-2})\ln(|\mathcal{H}|)}{n}} + \sqrt{\frac{\ln(e/\delta)}{n}} \right).
$$

To obtain Lemma 3.1, we balance the statements above, by setting $N = c\theta_{i+1}^{-2}\ln(e/l_{j+1})$ for a constant $c > 0$ big enough. Then $\exp(-N\theta_{i+1}^2/c) \leq l_{j+1}/e$, and combining the statements using a simple union bound, yields:

$$
\sup_{f \in \mathcal{C}_{\mathcal{H}}(\Theta_i, L_j)} \left( \mathcal{L}_{\mathcal{D}}(f) - \mathcal{L}_{\mathbf{S}}^{\theta}(f) \right)
$$
$$
\leq c \left( \sqrt{l_{j+1}\left(\frac{\ln(e/l_{j+1})\ln(|\mathcal{H}|)}{\theta_{i+1}^2 n} + \frac{\ln(1/\delta)}{n}\right)} \right.
$$
$$
+ \frac{\ln(e/l_{j+1})\ln(|\mathcal{H}|)}{\theta_{i+1}^2 n} + \frac{\ln(1/\delta)}{n} \right)
$$
$$
+ c \left( l_{j+1}\sqrt{\frac{\ln(|\mathcal{H}|)}{n\theta_{i+1}^2}}(\ln(e/l_{j+1}) + 1) + l_{j+1}\sqrt{\frac{\ln(e/\delta)}{n}} \right).
$$

Which yields Lemma 3.1, and thus finishes the proof of our main result, Theorem 1.1.

## 4. Conclusion

We have proven the first asymptotically tight margin-based generalization bound for voting classifiers over finite hypothesis sets. This finally settles the decades long discussion on

the generalization performance of voting classifiers, started by (Schapire et al., 1998). Our main theorem explicitly describes the generalizability of voting classifiers in terms of the size of the hypothesis set, the margin, the fraction of training points with the given margin, the number of training samples and the failure probability.

We reiterate the thoughts of (Larsen & Schalburg, 2025), where our techniques are adopted from: This updated version of the framework of (Schapire et al., 1998) could be applicable to other related settings with similar gaps in their generalization bounds. In this paper we have clarified the structure of the updated framework; combined with the natural cohesion between the framework and the setting of voting classifiers, we have improved adaptability. Lastly, we confirm the suspicion of (Larsen & Schalburg, 2025): indeed their techniques are adaptable in the setting of voting classifiers, culminating in the result of this paper.

## Impact Statement

This paper presents work whose goal is to advance the field of Machine Learning. There are many potential societal consequences of our work, none which we feel must be specifically highlighted here.

## Acknowledgment

This work is funded by the European Union (ERC, TUCLA, 101125203). Views and opinions expressed are however those of the author(s) only and do not necessarily reflect those of the European Union or the European Research Council. Neither the European Union nor the granting authority can be held responsible for them.

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

## A. Rademacher Bounds

In this section we proceed to use Rademacher complexity to bound (14) and (15). The bound is stated in the following lemma

**Restatement of Lemma 3.6.** *There exists a constant $c > 0$ such that when $N \geq 32\theta_{i+1}^{-2}$ it holds with probability at least $1 - \delta$ over the sample $\mathbf{S} \sim \mathcal{D}^n$ that:*

$$\max\{(14), (15)\} \leq c \exp\left(-\frac{\theta_{i+1}^2 N}{c}\right) \left(\sqrt{\frac{(\theta_{i+1}^2 N^2 + \theta_{i+1}^{-2})\ln(|\mathcal{H}|)}{n}} + \sqrt{\frac{\ln(e/\delta)}{n}}\right).$$

*Proof.* As the proof of for (15) is the same, we will only prove the bound for (14). Let $\sigma \sim \mathcal{R}^n$, denote a vector of independent random variables uniformly in $\{-1, +1\}$. Then the empirical Rademacher complexity of $\phi$ with respect to a training sample $S \in (\mathcal{X} \times \{-1, +1\})^n$ is:

$$\hat{\mathcal{R}}_{\phi, \mathcal{C}(\mathcal{H})}(S) = \frac{1}{n} \mathop{\mathbb{E}}_{\sigma \sim \mathcal{R}^n} \left[\sup_{f \in \mathcal{C}(\mathcal{H})} \sum_{i \in [n]} \sigma_i \phi(y_i f(x_i))\right].$$

As we will argue later, $\phi$ is $L_\phi$-Lipschitz, hence using the contraction inequality of (Ledoux & Talagrand, 1991), gives:

$$\hat{\mathcal{R}}_{\phi, \mathcal{C}(\mathcal{H})}(S) \leq \frac{L_\phi}{n} \mathop{\mathbb{E}}_{\sigma \sim \mathcal{R}^n} \left[\sup_{f \in \mathcal{C}(\mathcal{H})} \sum_{i \in [n]} \sigma_i y_i f(x_i)\right] = \frac{L_\phi}{n} \mathop{\mathbb{E}}_{\sigma \sim \mathcal{R}^n} \left[\sup_{f \in \mathcal{H}} \sum_{i \in [n]} \sigma_i f(x_i)\right] = \frac{L_\phi}{n} \hat{\mathcal{R}}_{\mathcal{H}}(S).$$

Where the second equality holds since $\sigma_i \overset{d}{=} \sigma_i y_i$ for all $i \in [n]$, as well as $\sup_{f \in \mathcal{C}(\mathcal{H})} \sum_{i \in [n]} \sigma_i f(x_i) = \sup_{f \in \mathcal{H}} \sum_{i \in [n]} \sigma_i f(x_i)$ The $\geq$ part clearly holds since $\mathcal{H} \subseteq \mathcal{C}(\mathcal{H})$. For the other way, we remark that for any $f \in \mathcal{C}(\mathcal{H})$, it holds that:

$$\sum_{i \in [n]} \sigma_i f(x_i) = \sum_{i \in [n]} \sigma_i \sum_{h \in \mathcal{H}} a_h h(x_i) = \sum_{h \in \mathcal{H}} a_h \sum_{i \in [n]} \sigma_i h(x_i) \leq \sup_{h \in \mathcal{H}} \sum_{i \in [n]} \sigma_i h(x_i)$$

meaning it also holds for the supremum over $\mathcal{C}(\mathcal{H})$

Now, since none of the steps above depend on $S$, it holds in expectation over $\mathbf{S} \sim \mathcal{D}$, i.e for Rademacher complexity in general:

$$\mathcal{R}_{\mathcal{D}, \phi, \mathcal{C}(\mathcal{H})}(n) \leq \frac{L_\phi}{n} \mathcal{R}_{\mathcal{D}, \mathcal{H}}(n) \leq c \frac{L_\phi}{\sqrt{n}} \sqrt{\ln(|\mathcal{H}|)}$$

Where the last inequality holds by (Massart, 2000) finite class lemma.

Now by Lemma D.2 and definition of $\phi$, it holds that:

$$0 \leq \phi(\lambda) \leq \max_{-c_\theta \leq \lambda \leq 0} \mathop{\mathbb{P}}_{\mathbf{g} \sim \mathcal{Q}_f} [y\mathbf{g}(x) > \theta_i/2 \mid yf(x) = \lambda] = \mathop{\mathbb{P}}_{\mathbf{g} \sim \mathcal{Q}_f} [y\mathbf{g}(x) > \theta_i/2 \mid yf(x) = 0]$$

Which by Lemma D.3 with $k^* = \lfloor (\frac{\theta_i}{4} + \frac{1}{2})N \rfloor + 1$ and Hoeffding, is bounded by:

$$\mathop{\mathbb{P}}_{H \sim \text{Binom}(N, 1/2)} [H - \mathbb{E}[H] \geq k^* - N/2] \leq \exp\left(-\frac{2(k^* - \frac{N}{2})^2}{N}\right) \leq \exp\left(-\frac{N\theta_i^2}{16}\right)$$

Using that $k^* - N/2 > N\theta_i/4$.

Finally, by standard generalization bounds using Rademacher complexity (see for example (Shalev-Shwartz & Ben-David, 2014)), it holds with probability $1 - \delta$, over $\mathbf{S} \sim \mathcal{D}^n$ that:

$$\sup_{f \in \mathcal{C}} \left| \mathop{\mathbb{E}}_{(\mathbf{x}, \mathbf{y}) \sim \mathcal{D}} [\phi(\mathbf{y}f(\mathbf{x}))] - \mathop{\mathbb{E}}_{(\mathbf{x}, \mathbf{y}) \sim S} [\phi(\mathbf{y}f(\mathbf{x}))] \right| \leq 2\mathcal{R}_{\mathcal{D}, \phi, \mathcal{C}(\mathcal{H})}(n) + \exp\left(-\frac{N\theta_i^2}{16}\right) c' \sqrt{\frac{\ln(1/\delta)}{n}}$$

$$\leq 2c \frac{L_\phi}{\sqrt{n}} \sqrt{\ln(|\mathcal{H}|)} + \exp\left(-\frac{N\theta_i^2}{16}\right) c' \sqrt{\frac{\ln(1/\delta)}{n}}$$

Remembering that $2\theta_i = \theta_{i+1}$ and combining with a bound of the Lipschitz constant $L_\phi$, as described in the following lemma:

**Lemma A.1.** *There exists a constant $c > 0$ such that the Lipschitz constants $L_\phi$, $L_\rho$ of $\phi$ and $\rho$ are bounded by*

$$c \exp\left(-\frac{N\theta_{i+1}^2}{c}\right)(\theta_{i+1}N + \theta_{i+1}^{-1})$$

*when $N \geq 32 \cdot \theta_{i+1}^{-2}$.*

This finishes the proof. $\square$

In the next section, we prove Lemma A.1.

### A.1. Bounding the Lipschitz Constants

As $\phi$ and $\rho$ are piecewise functions, their Lipshitz constants are bounded by the sum of the Lipshitz constants for the piece functions. Hence to prove Lemma A.1, we need to prove the following 2 lemmas:

**Lemma A.2.** *There exists a constant $c > 0$ such that the Lipschitz constant of $\phi$ for $0 < \lambda \leq \theta$ and $\rho$ for $0 < \lambda \leq \theta$ is bounded by:*

$$c\theta_{i+1}^{-1} \exp\left(-\frac{N\theta_{i+1}^2}{c}\right).$$

**Lemma A.3.** *There exists a constant $c > 0$ such that the Lipschitz constant of $\phi$ for $-1/\sqrt{2} \leq \lambda \leq 0$ and $\rho$ for $\theta_i < \lambda \leq 1/\sqrt{2}$ is bounded by:*

$$cN\theta_{i+1} \exp\left(-\frac{N\theta_{i+1}^2}{c}\right)$$

*when $N \geq 32 \cdot \theta_{i+1}^{-2}$.*

Remark: in Subsection 3.1 we argued that we can reduce to margins in $[-c_\theta, c_\theta]$ for a constant $0 < c_\theta < 1$, we have now made use of this reduction, and chosen $c_\theta = 1/\sqrt{2}$.

The easiest is Lemma A.2, which we start with:

*Proof of lemma A.2.* Firstly, when $0 < \lambda \leq \theta_i$, $\phi(\lambda)$ and $\rho(\lambda)$ are defined as

$$\phi(\lambda) = \frac{\theta_i - \lambda}{\theta_i} \mathop{\mathbb{P}}_{\mathbf{g} \sim \mathcal{Q}_f}[y\mathbf{g}(x) > \theta_i/2 \mid yf(x) = 0] \qquad \text{and} \qquad \rho(\lambda) = \frac{\lambda}{\theta_i} \mathop{\mathbb{P}}_{\mathbf{g} \sim \mathcal{Q}_f}[y\mathbf{g}(x) > \theta_i/2 \mid yf(x) = \theta_i].$$

Neither probability depends on the value of $\lambda$, hence the Lipschitz constants are bounded by $1/\theta_i$ times the probability. Both probabilities are by Lemma D.2, Lemma D.3 and the Hoeffding bound bounded by.

$$\mathop{\mathbb{P}}_{\mathbf{g} \sim \mathcal{Q}_f}[y\mathbf{g}(x) > \theta_i/2 \mid yf(x) = \theta_i] \leq \exp\left(-\frac{\theta_i^2 N}{8}\right) \leq \exp\left(-\frac{\theta_{i+1}^2 N}{32}\right)$$

as wanted $\square$

*Proof of Lemma A.3.* We will go in detail with the argument for $\phi$ when $-1/\sqrt{2} \leq \lambda \leq 0$, and remark when the argument for $\rho$ with $\theta < \lambda \leq 1/\sqrt{2}$ is different.

For $\phi$ with $-1/\sqrt{2} \leq \lambda \leq 0$, we need to bound the Lipschitz constant of:

$$\mathop{\mathbb{P}}_{\mathbf{g} \sim \mathcal{Q}_f}[y\mathbf{g}(x) > \theta_i/2 \mid yf(x) = \lambda].$$

By Lemma D.3, this probability equals:

$$\underset{\mathbf{H}\sim\text{Binom}(N,p_h)}{\mathbb{P}}[\mathbf{H}\geq k^*] = \sum_{k=k^*}^{N}\binom{N}{k}p_h^k(1-p_h)^{N-k}$$

where $p_h = \frac{1}{2} + \frac{\lambda}{2}$ and $k^* = \lfloor(\frac{\theta_i}{4} + \frac{1}{2})N\rfloor + 1$.

To bound the Lipschitz constant, we bound the absolute differential:

$$\left|\frac{\partial}{\partial\lambda}\underset{\mathbf{g}\sim\mathcal{Q}_f}{\mathbb{P}}[y\mathbf{g}(x) > \theta_i/2 \mid yf(x) = \lambda]\right| = \frac{1}{2}\sum_{k=k^*}^{N}\binom{N}{k}p_h^{k-1}(1-p_h)^{N-k-1}|k-p_hN|$$

$$= \frac{1}{2p_h(1-p_h)}\sum_{k=k^*}^{N}\binom{N}{k}p_h^k(1-p_h)^{N-k}|k-p_hN|$$

$$= \frac{1}{8(1-\lambda^2)}\sum_{k=k^*}^{N}\binom{N}{k}p_h^k(1-p_h)^{N-k}|k-p_hN|.$$

The same analysis gives the same expression for $\rho$, since it is just the converse probability. By choice of $c_\theta$ we have $(1-\lambda^2) \geq 1/2$.

Let now $\Delta = k^* - p_hN$, and define $I_0 = [k^*, k^*+2\Delta]$ and $I_j = [k^*+2^j\Delta, k^*+2^{j+1}\Delta]$ for $j = 1,\ldots,j^*$ where $j^*$ is the smallest integer such that $k^* + 2^{j^*+1}\Delta \geq N$.

Then the sum above can be split up according to $I_j$, and we get an upper bound:

$$\frac{1}{16}\sum_{j=0}^{j^*}\sum_{k\in I_j}\binom{N}{k}p_h^k(1-p_h)^{N-k}|k-p_hN|. \tag{16}$$

Then for any $k \in I_j$ with $j = 1,\ldots,j^*$ it holds that:

$$(2^j+1)\Delta = k^*+2^j\Delta - p_hN \leq k - p_hN \leq k^*+2^{j+1}\Delta - p_hN = (2^{j+1}+1)\Delta$$

i.e., it holds that: $|k-p_hN| \leq \max\{(2^j+1)|\Delta|, (2^{j+1}+1)|\Delta|\} = (2^{j+1}+1)|\Delta|$ as well as for $j = 0$.

Returning to (16) we obtain the upper bound for $\phi$ when $\lambda \in [-c_\theta, 0]$ and $\rho$ for $\lambda \in (\theta, c_\theta]$:

$$\frac{1}{16}|\Delta|\sum_{j=0}^{j^*}(2^{j+1}+1)\sum_{k\in I_j}\binom{N}{k}p_h^k(1-p_h)^{N-k}.$$

Replacing the innermost sum with the corresponding probability, gives an upper bound:

$$\frac{1}{16}|\Delta|\sum_{j=0}^{j^*}(2^{j+1}+1)\underset{\mathbf{H}\sim\text{Binom}(N,p_h)}{\mathbb{P}}[\mathbf{H}\in I_j].$$

We continue with the probability in the expression.

$$\underset{\mathbf{H}\sim\text{Binom}(N,p_h)}{\mathbb{P}}[\mathbf{H}\in I_j] = \underset{\mathbf{H}\sim\text{Binom}(N,p_h)}{\mathbb{P}}[k^*+2^j\Delta \leq \mathbf{H} \leq k^*+2^{j+1}\Delta]$$

$$= \underset{\mathbf{H}\sim\text{Binom}(N,p_h)}{\mathbb{P}}[(2^j+1)\Delta \leq \mathbf{H}-\mathbb{E}[\mathbf{H}] \leq (2^{j+1}+1)\Delta].$$

Applying Hoeffding on the above, dependending on the sign of $\Delta$:

if $\Delta > 0$ then $\underset{\mathbf{H}\sim\text{Binom}(N,p_h)}{\mathbb{P}}[\mathbf{H}\in I_j] \leq \underset{\mathbf{H}\sim\text{Binom}(N,p_h)}{\mathbb{P}}[(2^j+1)\Delta \leq \mathbf{H}-\mathbb{E}[\mathbf{H}]] < \exp\left(-\frac{2(2^j+1)^2\Delta^2}{N}\right).$

if $\Delta < 0$ then $\underset{\mathbf{H}\sim\text{Binom}(N,p_h)}{\mathbb{P}}[\mathbf{H}\in I_j] \leq \underset{\mathbf{H}\sim\text{Binom}(N,p_h)}{\mathbb{P}}[\mathbf{H}-\mathbb{E}[\mathbf{H}] \leq -(2^{j+1}+1)|\Delta|] < \exp\left(-\frac{2(2^{j+1}+1)^2|\Delta|^2}{N}\right).$

Hence for both cases, we bound the probability with the largest of the above.

$$\mathbb{P}_{\mathbf{H}\sim\mathrm{Binom}(N,p_h)}[\mathbf{H} \in I_j] < \exp\left(-\frac{2(2^j+1)^2|\Delta|^2}{N}\right).$$

To summarize, we have so far argued that:

$$|\frac{\partial}{\partial\lambda}\mathbb{P}_{\mathbf{g}\sim\mathcal{Q}_f}[y\mathbf{g}(x) > \theta_i/2 \mid yf(x) = \lambda]| < \frac{1}{16}|\Delta|\sum_{j=0}^{j^*}(2^{j+1}+1)\exp\left(-\frac{2(2^j+1)^2|\Delta|^2}{N}\right). \tag{17}$$

Covering both $\phi$ when $\lambda \in [-c_\theta, 0]$ and $\rho$ when $\lambda \in (\theta_i, c_\theta]$.

Now we examine this bound when $\theta_i$ behaves nicely. For $\phi$, the analysis is easiest for $-\theta_i \le \lambda \le 0$. Remark that the assumption $N \ge 32 \cdot \theta_{i+1}^{-2}$ implies $N\theta_i \ge 1$ and $N\theta_i/4 \ge 1$.

$$\lambda \le 0 \implies \Delta = \lfloor \frac{N}{2} + \frac{N\theta_i}{4} \rfloor + 1 - \frac{N}{2} - \frac{N\lambda}{2} > \frac{N\theta_i}{4} - \frac{N\lambda}{2} \ge \frac{N\theta_i}{4} > 0$$

$$-\theta_i \le \lambda \le 0 \implies \Delta \le \frac{N\theta_i}{4} + 1 - \frac{N\lambda}{2} \le \frac{N\theta_i}{2} + 1 + \frac{N\theta_i}{2} \le 2N\theta_i \qquad \text{since } N\theta_i \ge 1$$

For $\rho$ the analysis is easiest when $\theta_i \le \lambda \le (9/4)\theta_i$.

$$\theta_i \le \lambda \le (9/4)\theta_i \implies \Delta > \frac{N\theta_i}{4} - \frac{N\lambda}{2} \ge \frac{N\theta_i}{4} - \frac{9N\theta_i}{4} = -2N\theta_i$$

$$\theta_i \le \lambda \implies \Delta \le \frac{N\theta_i}{4} + 1 - \frac{N\lambda}{2} \le \frac{N\theta_i}{4} + 1 - \frac{N\theta_i}{2} = -\frac{N\theta_i}{2} + 1 \le -\frac{N\theta_i}{4} \qquad \text{since } N\theta_i/4 \ge 1$$

Hence for both nice cases, $|\Delta| \le 2N\theta_i$ and $|\Delta|^2 = \Delta^2 \ge N^2\theta_i^2/16$, reducing the upper bound to:

$$\frac{1}{8}N\theta_i\sum_{j=0}^{j^*}(2^{j+1}+1)\exp\left(-\frac{2(2^j+1)^2N\theta_i^2}{16}\right) \le \frac{1}{8}N\theta_i\sum_{j=0}^{j^*}\exp(2^{j+1})\exp\left(-\frac{(2^j+1)^2N\theta_i^2}{8}\right)$$

Now the assumption also implies $N\theta_i^2 \ge 8$, which implies that $8N \cdot 2^{j+1} \le 2^{j+1}N\theta_i^2$ and $\exp(-N\theta_i^2/8) \le 1/2$, hence the bound becomes:

$$\frac{1}{8}N\theta_i\sum_{j=0}^{j^*}\exp\left(-\frac{(2^{2j}+1)N\theta_i^2}{8}\right) = \frac{1}{8}N\theta_i\sum_{j=0}^{j^*}\exp\left(-\frac{N\theta_i^2}{8}\right)^{2^{2j}+1}$$

$$\le \frac{1}{8}N\theta_i\exp\left(-\frac{N\theta_i^2}{8}\right)\sum_{j=0}^{\infty}\exp\left(-\frac{N\theta_i^2}{8}\right)^j \le N\theta_i\exp\left(-\frac{N\theta_i^2}{8}\right)$$

Yielding the wanted bound for the nice cases, since $2\theta_i = \theta_{i+1}$.

For the not-nice cases, we return to (17), and differentiate the right side with respect to $\lambda$. For the not-nice cases to follow from the nice cases, we need the differential to be increasing for $\lambda \le -\theta_i$, and decreasing for $(9/4)\theta_i < \lambda$.

We start with $\lambda \le -\theta_i$. Since $\Delta > 0$, we can reduce to differentiating (17) with respect to $\Delta$.

$$\frac{1}{16}\sum_{j=0}^{j^*}(2^{j+1}+1)\frac{\partial}{\partial\Delta}\left(\Delta\exp\left(-\frac{2(2^j+1)^2\Delta^2}{N}\right)\right)$$

$$= \frac{1}{16}\sum_{j=0}^{j^*}(2^{j+1}+1)\left(\exp\left(-\frac{2(2^j+1)^2\Delta^2}{N}\right) + \Delta\exp\left(-\frac{2(2^j+1)^2\Delta^2}{N}\right)\cdot\left(-\frac{2(2^j+1)^2}{N}\right)\cdot 2\Delta\right).$$

We remark that since $\Delta$ is decreasing in $\lambda$, we want this differential to be negative, since then in total it will be increasing in $\lambda$.

The expression above is negative for all $j = 0, \ldots, j^*$, if and only if

$$1 - \frac{(2^j + 1)^2}{N} 4\Delta^2 < 0 \qquad \Longleftrightarrow \qquad N < 4(2^j + 1)^2 \Delta^2$$

This is indeed true, since $\Delta^2 > N^2 \theta_i^2 / 16$ and $N\theta_i^2 \geq 8$. Finishing the not nice case for $\phi$

We continue with $(9/4)\theta_i < \lambda$, here we remark that: $\Delta > \frac{N\theta_i}{4} - \frac{N\lambda}{2} > -\frac{N\lambda}{2}$ making $|\Delta| \leq \max\{|-\frac{N\theta_i}{4}|, |-\frac{N\lambda}{2}|\} = \frac{N\lambda}{2}$ so now we can remove the absolute value and differentiate:

$$\frac{N}{32} \sum_{j=0}^{j^*} (2^{j+1} + 1) \frac{\partial}{\partial \lambda} \left( \lambda \exp\left( -\frac{2(2^j + 1)^2 \Delta^2}{N} \right) \right)$$

$$= \frac{N}{32} \sum_{j=0}^{j^*} (2^{j+1} + 1) \left( \exp\left( -\frac{2(2^j + 1)^2 \Delta^2}{N} \right) + \lambda \exp\left( -\frac{2(2^j + 1)^2 \Delta^2}{N} \right) \cdot \left( -\frac{2(2^j + 1)^2}{N} \right) \cdot 2\Delta \left( -\frac{N}{2} \right) \right)$$

this differential is negative for all $j = 0, \ldots, j^*$, if and only if

$$1 + \lambda(2^j + 1)^2 2\Delta < 0 \qquad \Longleftrightarrow \qquad -\lambda(2^j + 1)^2 2\Delta > 1$$

Firstly, using the not-nice assumption $\lambda > (9/4)\theta_i$, and $N\theta_i^2 \geq 8$

$$2\Delta \leq \frac{N\theta_i}{2} + 2 - N\lambda < \frac{N\theta_i}{2} + 2 - \frac{9N\theta_i}{4} = 2 - \frac{7N\theta_i}{4} \leq -\frac{6N\theta_i}{4}$$

Finally,

$$-\lambda(2^j + 1)^2 2\Delta > \lambda(2^j + 1)^2 \frac{3}{2} N\theta_i \geq (2^j + 1)^2 \frac{3}{2} N\theta_i^2 \geq 12$$

Finishing the not-nice case for $\rho$, and thus finishing the proof. $\qquad\square$

## B. Half-Margin Generalization Bound

In this section, we proceed to bound (13) as stated in the following lemma:

**Restatement of Lemma 3.5.** *There exists $c > 0$ such that with probability greater than $1 - \delta$ over the sample $\mathbf{S} \sim \mathcal{D}^n$ it holds that:*

$$\sup_{f \in \mathcal{C}_{\mathcal{H}}(\Theta_i, L_j)} \left( \left| \mathbb{E}_{\mathbf{g} \sim \mathcal{Q}_f} [\mathcal{L}_{\mathcal{D}}^{\theta_i/2}(\mathbf{g}) - \mathcal{L}_{\mathbf{S}}^{\theta_i/2}(\mathbf{g})] \right| \right) \leq c \left( \sqrt{\frac{(l_{j+1} + \exp(-N\theta_{i+1}^2/c)) \ln(|\mathcal{C}_N|/\delta)}{n}} + \frac{\ln(|\mathcal{C}_N|/\delta)}{n} \right)$$

*When $N \geq 32 \cdot \theta_{i+1}^{-2}$*

*Proof.* Assume $N \geq 32\theta_{i+1}^{-2}$ and $c \geq \sqrt{3}$. Firstly, by Jensen's inequality, to prove the Lemma it's enough to prove that, with probability greater than $1 - \delta$ over the sample $\mathbf{S} \sim \mathcal{D}^n$, it holds that:

$$\sup_{f \in \mathcal{C}_{\mathcal{H}}(\Theta_i, L_j)} \left( \mathbb{E}_{\mathbf{g} \sim \mathcal{Q}_f} [|\mathcal{L}_{\mathcal{D}}^{\theta_i/2}(\mathbf{g}) - \mathcal{L}_{\mathbf{S}}^{\theta_i/2}(\mathbf{g})|] \right) \leq c \left( \sqrt{\frac{(l_{j+1} + \exp(-N\theta_{i+1}^2/c)) \ln(|\mathcal{H}|^N/\delta)}{n}} + \frac{\ln(|\mathcal{H}|^N/\delta)}{n} \right)$$

We start by fixing $g \in \mathcal{C}_N$ and remarking that:

$$\mathcal{L}_{\mathbf{S}}^{\theta_i/2}(g) = \mathbb{P}_{(\mathbf{x}, \mathbf{y}) \sim S}[\mathbf{y}g(\mathbf{x}) \leq \theta_i/2] = \frac{1}{n} \sum_{i \in [n]} \mathbb{1}_{\{y_i g(x_i) \leq \theta_i/2\}}$$

and additionally, $\mathbb{E}_{\mathbf{S} \sim \mathcal{D}^n}[\mathcal{L}_{\mathbf{S}}^{\theta_i/2}(g)] = \mathcal{L}_{\mathcal{D}}^{\theta_i/2}(g)$, hence using chernoff with

$$\varepsilon_N = \frac{c}{n} \left( \sqrt{\mathcal{L}_{\mathcal{D}}^{\theta_i/2}(g) n \ln(|\mathcal{C}_N|/\delta)} + \ln(|\mathcal{C}_N|/\delta) \right)$$

yields the bound:

$$
\mathbb{P}_{\mathbf{S}\sim\mathcal{D}^n}\big[\big|\mathcal{L}_{\mathcal{D}}^{\theta_i/2}(g) - \mathcal{L}_{\mathbf{S}}^{\theta_i/2}(g)\big| \geq \varepsilon_N\big] \leq 2\exp\left(-\frac{\varepsilon_N^2 n}{3\mathcal{L}_{\mathcal{D}}^{\theta_i/2}(g)}\right)
$$

Now since:

$$
\begin{aligned}
\varepsilon_N^2 = \frac{c^2}{n^2}\left(\sqrt{\mathcal{L}_{\mathcal{D}}^{\theta_i/2}(g)n\ln(|\mathcal{C}_N|/\delta)} + \ln(|\mathcal{C}_N|/\delta)\right)^2 &\geq \frac{c^2}{n^2}\max\{\mathcal{L}_{\mathcal{D}}^{\theta_i/2}(g)n\ln(|\mathcal{C}_N|/\delta), \ln(|\mathcal{C}_N|/\delta)^2\} \\
&= \frac{c^2}{n^2}\max\{\mathcal{L}_{\mathcal{D}}^{\theta_i/2}(g)n, \ln(|\mathcal{C}_N|/\delta)\}\ln(|\mathcal{C}_N|/\delta) \\
&\geq \frac{c^2}{n^2}\mathcal{L}_{\mathcal{D}}^{\theta_i/2}(g)n\ln(|\mathcal{C}_N|/\delta)
\end{aligned}
$$

We end up with a bound:

$$
\begin{aligned}
\mathbb{P}_{\mathbf{S}\sim\mathcal{D}^n}&\left[\left|\mathcal{L}_{\mathcal{D}}^{\theta_i/2}(g) - \mathcal{L}_{\mathbf{S}}^{\theta_i/2}(g)\right| \geq c\left(\sqrt{\frac{\mathcal{L}_{\mathcal{D}}^{\theta_i/2}(g)n\ln(|\mathcal{C}_N|/\delta)}{n}} + \frac{\ln(|\mathcal{C}_N|/\delta)}{n}\right)\right] \\
&\leq 2\exp\left(-\frac{(\frac{2}{n^2}\mathcal{L}_{\mathcal{D}}^{\theta_i/2}(g)n\ln(|\mathcal{C}_N|/\delta))n}{2\mathcal{L}_{\mathcal{D}}^{\theta_i/2}(g)}\right) \\
&= 2\exp\left(-\ln(|\mathcal{C}_N|/\delta)\right) = 2\delta/|\mathcal{C}_N|
\end{aligned}
$$

Halving $\delta$ and union bounding over $\mathcal{C}_N$, we get, with probability greater than $1 - \delta$ over the sample $\mathbf{S}\sim\mathcal{D}^n$ that:

$$
\forall g \in \mathcal{C}_N \qquad \left|\mathcal{L}_{\mathcal{D}}^{\theta_i/2}(g) - \mathcal{L}_{\mathbf{S}}^{\theta_i/2}(g)\right| \leq c\left(\sqrt{\frac{\mathcal{L}_{\mathcal{D}}^{\theta_i/2}(g)\ln(|\mathcal{C}_N|/\delta)}{n}} + \frac{\ln(|\mathcal{C}_N|/\delta)}{n}\right)
$$

Taking expectation over $\mathbf{g}\sim\mathcal{Q}_f$, the bound becomes:

$$
\mathbb{E}_{\mathbf{g}\sim\mathcal{Q}_f}\left[\left|\mathcal{L}_{\mathcal{D}}^{\theta_i/2}(\mathbf{g}) - \mathcal{L}_{\mathbf{S}}^{\theta_i/2}(\mathbf{g})\right|\right] \leq \mathbb{E}_{\mathbf{g}\sim\mathcal{Q}_f}\left[c\left(\sqrt{\frac{\mathcal{L}_{\mathcal{D}}^{\theta_i/2}(\mathbf{g})\ln(|\mathcal{C}_N|/\delta)}{n}} + \frac{\ln(|\mathcal{C}_N|/\delta)}{n}\right)\right]
$$

Since $f(x) = a\sqrt{x} + c$ is concave, Jensen's inequality yields:

$$
\mathbb{E}_{\mathbf{g}\sim\mathcal{Q}_f}\left[\left|\mathcal{L}_{\mathcal{D}}^{\theta_i/2}(\mathbf{g}) - \mathcal{L}_{\mathbf{S}}^{\theta_i/2}(\mathbf{g})\right|\right] \leq c\left(\sqrt{\frac{\mathbb{E}_{\mathbf{g}\sim\mathcal{Q}_f}\left[\mathcal{L}_{\mathcal{D}}^{\theta_i/2}(\mathbf{g})\right]\ln(|\mathcal{C}_N|/\delta)}{n}} + \frac{\ln(|\mathcal{C}_N|/\delta)}{n}\right)
$$

Now we can take $\sup_{f\in\mathcal{C}_{\mathcal{H}}(\Theta_i,L_j)}$ on both sides and drop it from the right side since it does not depend on $f$. To finish the proof, we just need to bound: $\mathbb{E}_{\mathbf{g}\sim\mathcal{Q}_f}[\mathcal{L}_{\mathcal{D}}^{\theta_i/2}(\mathbf{g})] \leq l_{j+1} + \exp(-N\theta_i^2/c)$.

Using the definition of margin loss and law of total expectation, we get:

$$
\begin{aligned}
\mathop{\mathbb{E}}_{\mathbf{g}\sim\mathcal{Q}_f}[\mathcal{L}_{\mathcal{D}}^{\theta_i/2}(\mathbf{g})] &= \mathop{\mathbb{E}}_{\mathbf{g}\sim\mathcal{Q}_f}\left[\mathop{\mathbb{P}}_{(\mathbf{x},\mathbf{y})\sim\mathcal{D}}\left[\mathbf{y}\mathbf{g}(\mathbf{x})\leq\frac{\theta_i}{2}\right]\right] \\
&= \mathop{\mathbb{E}}_{(\mathbf{x},\mathbf{y})\sim\mathcal{D}}\left[\mathop{\mathbb{P}}_{\mathbf{g}\sim\mathcal{Q}_f}\left[\mathbf{y}\mathbf{g}(\mathbf{x})\leq\frac{\theta_i}{2}\right]\right] \\
&= \mathop{\mathbb{E}}_{(\mathbf{x},\mathbf{y})\sim\mathcal{D}}\left[\mathop{\mathbb{P}}_{\mathbf{g}\sim\mathcal{Q}_f}\left[\mathbf{y}\mathbf{g}(\mathbf{x})\leq\frac{\theta_i}{2}\right]\ \middle|\ \mathbf{y}f(\mathbf{x})>\frac{3}{4}\theta_i\right]\cdot\mathop{\mathbb{P}}_{(\mathbf{x},\mathbf{y})\sim\mathcal{D}}\left[\mathbf{y}f(\mathbf{x})>\frac{3}{4}\theta_i\right] \\
&\quad + \mathop{\mathbb{E}}_{(\mathbf{x},\mathbf{y})\sim\mathcal{D}}\left[\mathop{\mathbb{P}}_{\mathbf{g}\sim\mathcal{Q}_f}\left[\mathbf{y}\mathbf{g}(\mathbf{x})\leq\frac{\theta_i}{2}\right]\ \middle|\ \mathbf{y}f(\mathbf{x})\leq\frac{3}{4}\theta_i\right]\cdot\mathcal{L}_{\mathcal{D}}^{(3/4)\theta_i}(f) \\
&\leq \mathop{\mathbb{E}}_{(\mathbf{x},\mathbf{y})\sim\mathcal{D}}\left[\mathop{\mathbb{P}}_{\mathbf{g}\sim\mathcal{Q}_f}\left[\mathbf{y}\mathbf{g}(\mathbf{x})\leq\frac{\theta_i}{2}\right]\ \middle|\ \mathbf{y}f(\mathbf{x})>\frac{3}{4}\theta_i\right]+\mathcal{L}_{\mathcal{D}}^{(3/4)\theta_i}(f)
\end{aligned}
$$

Now replacing the expectation with a supremum over margins and utilizing lemma D.2 and D.3,

$$
\begin{aligned}
\mathcal{L}_{\mathcal{D}}^{(3/4)\theta_i}(f) &+ \mathop{\mathbb{E}}_{(\mathbf{x},\mathbf{y})\sim\mathcal{D}}\left[\mathop{\mathbb{P}}_{\mathbf{g}\sim\mathcal{Q}_f}\left[\mathbf{y}\mathbf{g}(\mathbf{x})\leq\frac{\theta_i}{2}\right]\ \middle|\ \mathbf{y}f(\mathbf{x})>\frac{3}{4}\theta_i\right] \\
&\leq \mathcal{L}_{\mathcal{D}}^{(3/4)\theta_i}(f) + \sup_{\lambda>(3/4)\theta_i}\left(\mathop{\mathbb{P}}_{\mathbf{g}\sim\mathcal{Q}_f}\left[\mathbf{y}\mathbf{g}(\mathbf{x})\leq\frac{\theta_i}{2}\ \middle|\ \mathbf{y}f(\mathbf{x})=\lambda\right]\right) \\
&\leq \mathcal{L}_{\mathcal{D}}^{(3/4)\theta_i}(f) + \mathop{\mathbb{P}}_{\mathbf{g}\sim\mathcal{Q}_f}\left[\mathbf{y}\mathbf{g}(\mathbf{x})\leq\frac{\theta_i}{2}\ \middle|\ \mathbf{y}f(\mathbf{x})=\frac{3}{4}\theta_i\right] \\
&\leq \mathcal{L}_{\mathcal{D}}^{(3/4)\theta_i}(f) + \mathop{\mathbb{P}}_{\mathbf{H}\sim\mathrm{binom}(N,\frac{1}{2}+\frac{3}{8}\theta_i)}\left[\mathbf{H}\leq\lfloor(\frac{\theta_i}{4}+\frac{1}{2})N\rfloor+1\right] \\
&= \mathcal{L}_{\mathcal{D}}^{(3/4)\theta_i}(f) + \mathop{\mathbb{P}}_{\mathbf{H}\sim\mathrm{binom}(N,\frac{1}{2}+\frac{3}{8}\theta_i)}\left[\mathbf{H}-\mathbb{E}[\mathbf{H}]\leq\lfloor(\frac{\theta_i}{4}+\frac{1}{2})N\rfloor+1-\mathbb{E}[\mathbf{H}]\right]
\end{aligned}
$$

Remarking that $N\theta_i\geq16$ by assumption and $\mathbb{E}[\mathbf{H}]=\frac{N}{2}+\frac{3}{8}N\theta_i$:

$$
-\frac{N\theta_i}{8}=\frac{N\theta_i}{4}-\frac{3}{8}N\theta_i<\lfloor(\frac{\theta_i}{4}+\frac{1}{2})N\rfloor+1-\mathbb{E}[\mathbf{H}]\leq\frac{N\theta_i}{4}+1-\frac{3}{8}N\theta_i=1-\frac{N\theta_i}{8}\leq-\frac{N\theta_i}{16}\leq0
$$

Allows us to use the Hoeffding bound on the probability above, hence in total we archive the bound:

$$
\begin{aligned}
\mathop{\mathbb{E}}_{\mathbf{g}\sim\mathcal{Q}_f}[\mathcal{L}_{\mathcal{D}}^{\theta_i/2}(\mathbf{g})] &< \mathcal{L}_{\mathcal{D}}^{(3/4)\theta_i}(f) + \exp\left(-\frac{2(\lfloor(\frac{\theta_i}{4}+\frac{1}{2})N\rfloor+1-\mathbb{E}[\mathbf{H}])^2}{N}\right) \\
&\leq \mathcal{L}_{\mathcal{D}}^{(3/4)\theta_i}(f) + \exp\left(-\frac{2N^2\theta_i^2}{256N}\right) \\
&\leq \mathcal{L}_{\mathcal{D}}^{(3/4)\theta_i}(f) + \exp\left(-\frac{N\theta_i^2}{128}\right)
\end{aligned}
$$

which finishes the proof, since by definition $2\theta_i=\theta_{i+1}$, $\mathcal{L}_{\mathcal{D}}^{(3/4)\theta_i}(f)\leq l_{j+1}$ and $|\mathcal{C}_N|=|\mathcal{H}|^N$ $\qquad\square$

# C. Reducing to Smaller Tasks

In this section, we prove the following claim:

**Restatement of Lemma 3.2.** *There is a constant $c>0$, such that for any $0<\delta<1$ and any $\Theta_i=(\theta_i,\theta_{i+1}]$, it holds with probability at least $1-\delta$ over the sample $\mathbf{S}\sim\mathcal{D}^n$*

$$
\forall f\in\mathcal{H}:\qquad\frac{\mathcal{L}_{\mathcal{D}}^{(3/4)\theta_i}(f)}{2}-\mathcal{L}_{\mathbf{S}}^{\theta_i}(f)\leq c\left(\frac{\ln(\theta_{i+1}^2n/\ln(|\mathcal{H}|))\ln(|\mathcal{H}|)}{\theta_{i+1}^2n}+\frac{\ln(e/\delta)}{n}\right)
$$

*When $N\geq64\cdot\theta_{i+1}^{-2}$*

which together with Claim 3.3 allows us to reduce the main theorem into smaller tasks.

*Proof.* Firstly fix $g \in \mathcal{C}_N$, then we remark that:

$$\mathcal{L}_{\mathbf{S}}^{(7/8)\theta_i}(g) - \mathop{\mathbb{P}}_{(\mathbf{x},\mathbf{y})\sim S}\left[\mathbf{y}f(\mathbf{x}) > \theta_i \wedge \mathbf{y}g(\mathbf{x}) \le \frac{7}{8}\theta_i\right] = \mathop{\mathbb{P}}_{(\mathbf{x},\mathbf{y})\sim S}\left[\mathbf{y}f(\mathbf{x}) \le \theta_i \wedge \mathbf{y}g(\mathbf{x}) \le \frac{7}{8}\theta_i\right] \le \mathcal{L}_{\mathbf{S}}^{\theta_i}(f)$$

$$\mathcal{L}_{\mathcal{D}}^{(3/4)\theta_i}(f) - \mathop{\mathbb{P}}_{(\mathbf{x},\mathbf{y})\sim\mathcal{D}}\left[\mathbf{y}f(\mathbf{x}) \le \frac{3}{4}\theta_i \wedge \mathbf{y}g(\mathbf{x}) > \frac{7}{8}\theta_i\right] = \mathop{\mathbb{P}}_{(\mathbf{x},\mathbf{y})\sim\mathcal{D}}\left[\mathbf{y}f(\mathbf{x}) \le \frac{3}{4}\theta_i \wedge \mathbf{y}g(\mathbf{x}) \le \frac{7}{8}\theta_i\right] \le \mathcal{L}_{\mathbf{S}}^{(7/8)\theta_i}(g)$$

Hence we can bound the difference:

$$\frac{\mathcal{L}_{\mathcal{D}}^{(3/4)\theta_i}(f)}{2} - \mathcal{L}_{\mathbf{S}}^{\theta_i}(f) \le \frac{\mathcal{L}_{\mathbf{S}}^{(7/8)\theta_i}(g)}{2} - \mathcal{L}_{\mathbf{S}}^{(7/8)\theta_i}(g) \tag{18}$$

$$+ \mathop{\mathbb{P}}_{(\mathbf{x},\mathbf{y})\sim\mathcal{D}}\left[\mathbf{y}f(\mathbf{x}) \le \frac{3}{4}\theta_i \wedge \mathbf{y}g(\mathbf{x}) > \frac{7}{8}\theta_i\right]$$

$$+ \mathop{\mathbb{P}}_{(\mathbf{x},\mathbf{y})\sim S}\left[\mathbf{y}f(\mathbf{x}) > \theta_i \wedge \mathbf{y}g(\mathbf{x}) \le \frac{7}{8}\theta_i\right]$$

We bound the first term using Bernstein's inequality.

Just like in Section B, we use that:

$$\mathop{\mathbb{E}}_{\mathbf{S}\sim\mathcal{D}^n}[\mathcal{L}_{\mathbf{S}}^{(7/8)\theta_i}(g)] = \mathcal{L}_{\mathcal{D}}^{(7/8)\theta_i}(g) \qquad \text{and} \qquad \mathcal{L}_{\mathbf{S}}^{(7/8)\theta_i}(g) = \mathop{\mathbb{P}}_{(\mathbf{x},\mathbf{y})\sim S}\left[\mathbf{y}g(\mathbf{x}) \le \frac{7}{8}\theta_i\right] = \frac{1}{n}\sum_{i\in[n]} \mathbb{1}_{\{\mathbf{y}_i g(\mathbf{x}_i) \le (7/8)\theta_i\}}$$

Hence

$$\mathop{\mathbb{P}}_{\mathbf{S}\sim\mathcal{D}^n}\left[\mathcal{L}_{\mathcal{D}}^{(7/8)\theta_i}(g) - \mathcal{L}_{\mathbf{S}}^{(7/8)\theta_i}(g) \ge \frac{t}{n}\right] \le \exp\left(-\frac{\frac{1}{2}t^2}{n\mathcal{L}_{\mathcal{D}}^{(7/8)\theta_i}(g) + \frac{1}{3}t}\right)$$

Now choosing $t$

$$t = n \cdot \left(\frac{\mathcal{L}_{\mathcal{D}}^{(7/8)\theta_i}(g)}{2} + \frac{Z}{n}\right) \qquad \text{with} \qquad Z = 32 \cdot \ln(1/\delta')$$

and remarking that

$$t^2 \ge n^2 \max\left(\left(\frac{\mathcal{L}_{\mathcal{D}}^{(7/8)\theta_i}(g)}{2}\right)^2, \left(\frac{Z}{n}\right)^2\right) \ge \frac{n^2}{4}\max\left(\mathcal{L}_{\mathcal{D}}^{(7/8)\theta_i}(g), \frac{Z}{n}\right)^2$$

As well as:

$$n\mathcal{L}_{\mathcal{D}}^{(7/8)\theta_i}(g) + \frac{1}{3}t \le n\left(2\mathcal{L}_{\mathcal{D}}^{(7/8)\theta_i}(g) + \frac{Z}{n}\right) \le 4n\max\left(\mathcal{L}_{\mathcal{D}}^{(7/8)\theta_i}(g), \frac{Z}{n}\right)$$

gives the bound:

$$\mathop{\mathbb{P}}_{\mathbf{S}\sim\mathcal{D}^n}\left[\frac{\mathcal{L}_{\mathcal{D}}^{(7/8)\theta_i}(g)}{2} - \mathcal{L}_{\mathbf{S}}^{(7/8)\theta_i}(g) \ge \frac{Z}{n}\right] \le \exp\left(-\frac{\frac{n^2}{8}\max\left(\mathcal{L}_{\mathcal{D}}^{(7/8)\theta_i}(g), \frac{Z}{n}\right)^2}{4n\max\left(\mathcal{L}_{\mathcal{D}}^{(7/8)\theta_i}(g), \frac{Z}{n}\right)}\right)$$

$$\le \exp\left(-\frac{Z}{32}\right) = \delta'$$

setting $\delta' = \delta/|\mathcal{C}_N| = \delta/|\mathcal{H}|^N$, and union bounding over $\mathcal{C}_N$, we get, with probability $1 - \delta$ over the sample $\mathbf{S} \sim \mathcal{D}^n$ that:

$$\forall g \in \mathcal{C}_n \qquad \frac{\mathcal{L}_{\mathcal{D}}^{(7/8)\theta_i}(g)}{2} - \mathcal{L}_{\mathbf{S}}^{(7/8)\theta_i}(g) \le 32\left(\frac{\ln(1/\delta)}{n} + \frac{\ln(|\mathcal{H}|)N}{n}\right)$$

This bounds one part of (18), for all $g \in \mathcal{C}_N$, i.e. it holds in expectation over $g \sim \mathcal{Q}_f$. This allows us to take the expectation over $g \sim \mathcal{Q}_f$, in (18)

$$\frac{\mathcal{L}_{\mathcal{D}}^{(3/4)\theta_i}(f)}{2} - \mathcal{L}_{\mathbf{S}}^{\theta_i}(f) \leq 32 \left( \frac{\ln(1/\delta)}{n} + \frac{\ln(|\mathcal{H}|)N}{n} \right) \tag{19}$$
$$+ \mathop{\mathbb{E}}_{g \sim \mathcal{Q}_f} \left[ \mathop{\mathbb{P}}_{(\mathbf{x},\mathbf{y}) \sim \mathcal{D}} \left[ \mathbf{y}f(\mathbf{x}) \leq \frac{3}{4}\theta_i \ \wedge \ \mathbf{y}g(\mathbf{x}) > \frac{7}{8}\theta_i \right] \right]$$
$$+ \mathop{\mathbb{E}}_{g \sim \mathcal{Q}_f} \left[ \mathop{\mathbb{P}}_{(\mathbf{x},\mathbf{y}) \sim S} \left[ \mathbf{y}f(\mathbf{x}) > \theta_i \ \wedge \ \mathbf{y}g(\mathbf{x}) \leq \frac{7}{8}\theta_i \right] \right]$$

Finally, we bound the last two terms, as the computations are similar, we will focus on boundin the first of these terms.

Just like section B, we replace the expectation with a supremum over margins and utilize lemma D.2 and D.3,

$$\mathop{\mathbb{E}}_{g \sim \mathcal{Q}_f} \left[ \mathop{\mathbb{P}}_{(\mathbf{x},\mathbf{y}) \sim S} \left[ \mathbf{y}f(\mathbf{x}) > \theta_i \ \wedge \ \mathbf{y}g(\mathbf{x}) \leq \frac{7}{8}\theta_i \right] \right] \leq \mathop{\mathbb{E}}_{g \sim \mathcal{Q}_f} \left[ \mathop{\mathbb{P}}_{(\mathbf{x},\mathbf{y}) \sim S} \left[ \mathbf{y}g(\mathbf{x}) \leq \frac{7}{8}\theta_i \ \middle| \ \mathbf{y}f(\mathbf{x}) > \theta_i \right] \right]$$
$$\leq \sup_{\lambda > \theta_i} \left( \mathop{\mathbb{P}}_{(\mathbf{x},\mathbf{y}) \sim S} \left[ \mathbf{y}g(\mathbf{x}) \leq \frac{7}{8}\theta_i \ \middle| \ \mathbf{y}f(\mathbf{x}) = \lambda \right] \right)$$
$$\leq \mathop{\mathbb{P}}_{(\mathbf{x},\mathbf{y}) \sim S} \left[ \mathbf{y}g(\mathbf{x}) \leq \frac{7}{8}\theta_i \ \middle| \ \mathbf{y}f(\mathbf{x}) = \theta_i \right]$$
$$= \mathop{\mathbb{P}}_{\mathbf{H} \sim \text{binom}(N, \frac{1}{2}+\frac{\theta_i}{2})} \left[ \mathbf{H} \leq \left\lfloor \left( \frac{7\theta_i}{16} + \frac{1}{2} \right) N \right\rfloor + 1 \right]$$
$$= \mathop{\mathbb{P}}_{\mathbf{H} \sim \text{binom}(N, \frac{1}{2}+\frac{\theta_i}{2})} \left[ \mathbf{H} - \mathbb{E}[\mathbf{H}] \leq \left\lfloor \left( \frac{7\theta_i}{16} + \frac{1}{2} \right) N \right\rfloor + 1 - \mathbb{E}[\mathbf{H}] \right]$$

Remarking that $N\theta_i \geq 32$ and $\mathbb{E}[\mathbf{H}] = \frac{N}{2} + \frac{1}{2}N\theta_i$:

$$-\frac{N\theta_i}{16} = \frac{7N\theta_i}{16} - \frac{N\theta_i}{2} < \left\lfloor \left( \frac{7\theta_i}{16} + \frac{1}{2} \right) N \right\rfloor + 1 - \mathbb{E}[\mathbf{H}] \leq \frac{7N\theta_i}{16} - \frac{N\theta_i}{2} + 1 = -\frac{N\theta_i}{16} + 1 \leq -\frac{N\theta_i}{32} \leq 0$$

Allows us to use the Hoeffding bound on the probability above, hence in total we archive the bound:

$$\mathop{\mathbb{E}}_{g \sim \mathcal{Q}_f} \left[ \mathop{\mathbb{P}}_{(\mathbf{x},\mathbf{y}) \sim S} \left[ \mathbf{y}f(\mathbf{x}) > \theta_i \ \wedge \ \mathbf{y}g(\mathbf{x}) \leq \frac{7}{8}\theta_i \right] \right] \leq \exp \left( -\frac{2(\lfloor \left( \frac{7\theta_i}{16} + \frac{1}{2} \right) N \rfloor + 1 - \mathbb{E}[\mathbf{H}])^2}{N} \right)$$
$$\leq \exp \left( -\frac{2N^2\theta_i^2}{2^{10}N} \right) = \exp \left( -\frac{N\theta_{i+1}^2}{2^{11}} \right)$$

Similar computation gives the same bound for the other term, including both in (19) yields:

$$\frac{\mathcal{L}_{\mathcal{D}}^{(3/4)\theta_i}(f)}{2} - \mathcal{L}_{\mathbf{S}}^{\theta_i}(f) \leq 32 \left( \frac{\ln(1/\delta)}{n} + \frac{\ln(|\mathcal{H}|)N}{n} \right) + 2\exp \left( -\frac{N\theta_{i+1}^2}{2^{11}} \right)$$

Choosing $N = 2^{11}\theta_{i+1}^{-2}\ln(\theta_{i+1}^2 n / \ln(|\mathcal{H}|))$ then yields:

$$\frac{\mathcal{L}_{\mathcal{D}}^{(3/4)\theta_i}(f)}{2} - \mathcal{L}_{\mathbf{S}}^{\theta_i}(f) \leq 32\frac{\ln(1/\delta)}{n} + 2^{15}\frac{\ln(|\mathcal{H}|)\ln(\theta_{i+1}^2 n / \ln(|\mathcal{H}|))}{\theta_{i+1}^2 n} + 2\frac{\ln(|\mathcal{H}|)}{\theta_{i+1}^2 n}$$

which gives the wanted result. $\qquad\square$

## D. Additional Proofs

### D.1. Reduction Claims

**Observation D.1.** *We can assume without loss of generality, that all margins lie in the interval: $[-c_\theta, c_\theta]$ for a constant $c_\theta \in (0,1)$*

*Proof.* Define hypotheses $h_+$ and $h_-$ to be the constant $+1$ and $-1$ hypotheses, then let $\overline{\mathcal{H}} = \mathcal{H} \cup \{h_+, h_-\}$ be the hypothesis set with these additions.

We want to map each voting classifier $f \in \mathcal{C}_{\mathcal{H}}$ to another voting classifier $\overline{f} \in \mathcal{C}_{\overline{\mathcal{H}}}$, such that: all $(\mathbf{x}, \mathbf{y}) \sim \mathcal{D}, \mathbf{y}f(\mathbf{x}) \in [-1, +1]$ implies that $\mathbf{y}\overline{f}(\mathbf{x}) \in [-c_\theta, c_\theta]$ and $\text{sign}(\overline{f}) = \text{sign}(f)$

Consider the function $\Phi$

$$\Phi : \mathcal{C}_{\mathcal{H}} \to \mathcal{C}_{\overline{\mathcal{H}}} \qquad \text{given by:} \qquad f \mapsto \overline{f} = c_\theta f + \frac{1 - c_\theta}{2}(h_+ + h_-)$$

Firstly, $\overline{f} \in \mathcal{C}_{\overline{\mathcal{H}}}$, since $f \in \mathcal{C}_{\mathcal{H}}$. Then, $h_+(x) + h_-(x) = 0$ for all $x$, implying the first condition. Lastly, $\text{sign}(h_+) = -\text{sign}(h_-)$, yields the last condition.

Hence by exchanging $\mathcal{H}$ with $\overline{\mathcal{H}}$, and mapping each $f$ to $\overline{f}$, we get smaller margins. This replacement is reversible and only changes the size of the hypothesis class by an additive constant. Hence we can make the exchange without losing generality. $\qquad\square$

**Restatement of Claim 3.3.** *For any $0 < \delta < 1$, the following holds:*

1. *If the sample $\mathbf{S} \sim \mathcal{D}^n$ satisfies (9) and (10) simultaneously for all $(\Theta_i, L_j)$ and $\Theta_i$, with slightly different constants, then for that sample, (8) holds for all margins $\theta \in \left(\sqrt{e\ln(|\mathcal{H}|)/n}, 1\right]$ and all $f \in \mathcal{C}(\mathcal{H})$.*

2. *With probability at least $1 - \delta$ over the sample $\mathbf{S} \sim \mathcal{D}^n$, the above event happens*

*Proof.* Let in the following $0 < \delta < 1$, $\Theta_i = (\theta_i, \theta_{i+1}]$ with $\theta_{i+1} = e2^i\sqrt{\ln|\mathcal{H}|/n}$ for $i = 1, \ldots, \log_2((e/c_\theta)\sqrt{n/\ln|\mathcal{H}|})$, $L_0 = [0, n^{-1}]$ and $L_j = (l_j, l_{j+1}]$, with $l_{j+1} = 2^j n^{-1}$ for $j = 1, \ldots, \log_2(n)$. We start by proving part 1.

We assume that $\mathbf{S} \sim \mathcal{D}^n$ is a sample such that for all $(\Theta_i, L_j)$ and $\Theta_i$, it holds that:

$$\sup_{\substack{f \in \mathcal{C}_{\mathcal{H}}(\Theta_i, L_j) \\ \theta \in \Theta_i}} |\mathcal{L}_{\mathcal{D}}(f) - \mathcal{L}_{\mathbf{S}}^{\theta}(f)| \le c_{3.1}\left(\sqrt{l_{j+1}\left(\frac{\ln(e/l_{j+1})\ln(|\mathcal{H}|)}{\theta_{i+1}^2 n} + \frac{\ln(e/\delta)}{n}\right)} + \frac{\ln(e/l_{j+1})\ln(|\mathcal{H}|)}{\theta_{i+1}^2 n} + \frac{\ln(e/\delta)}{n}\right)$$

$$(20)$$

and

$$\forall f \in \mathcal{C}_{\mathcal{H}} : \qquad \frac{\mathcal{L}_{\mathcal{D}}^{(3/4)\theta_i}(f)}{2} - \mathcal{L}_{\mathbf{S}}^{\theta_i}(f) \le c_{3.2}\left(\frac{\ln(\theta_{i+1}^2 n/\ln(|\mathcal{H}|))\ln(|\mathcal{H}|)}{\theta_{i+1}^2 n} + \frac{\ln(e/\delta)}{n}\right) \qquad (21)$$

Now fix $\theta \in \left(\sqrt{e\ln(|\mathcal{H}|)/n}, 1\right]$ and $f \in \mathcal{C}_{\mathcal{H}}$.

Then there exists $i, j$ such that $\theta \in \Theta_i = (\theta_i, \theta_{i+1}]$ and $\mathcal{L}_{\mathcal{D}}^{(3/4)\theta_i}(f) \in L_j = (l_j, l_{j+1}]$, where $\theta_{i+1} = e2^i\sqrt{\ln|\mathcal{H}|/n}$ and $l_{j+1} = 2^i n^{-1}$

We split into two cases, depending on whether or not it holds that:

$$\mathcal{L}_{\mathcal{D}}^{(3/4)\theta_i}(f) \le 8c_{3.2}\left(\frac{\ln(\theta_{i+1}^2 n/\ln(|\mathcal{H}|))\ln(|\mathcal{H}|)}{\theta_{i+1}^2 n} + \frac{\ln(e/\delta)}{n}\right)$$

Firstly, when this is true, we remark that: $\mathcal{L}_{\mathcal{D}}(f) \le \mathcal{L}_{\mathcal{D}}^{(3/4)\theta_i}(f)$, $\theta \le \theta_{i+1}$ and $(\ln(e\theta^2 n/\ln(|\mathcal{H}|))/\theta^2 n)$ is decreasing in $\theta \ge \sqrt{\ln(|\mathcal{H}|)/n}$, hence:

$$\mathcal{L}_{\mathcal{D}}(f) \le \mathcal{L}_{\mathcal{D}}^{(3/4)\theta_i}(f) \le 8c_{3.2}\left(\frac{\ln(\theta_{i+1}^2 n/\ln(|\mathcal{H}|))\ln(|\mathcal{H}|)}{\theta_{i+1}^2 n} + \frac{\ln(e/\delta)}{n}\right) \le 8c_{3.2}\left(\frac{\ln(\theta^2 n/\ln(|\mathcal{H}|))\ln(|\mathcal{H}|)}{\theta^2 n} + \frac{\ln(e/\delta)}{n}\right)$$

yielding (8) in this case.

For the other case, we remark that by definition of $j$, $\mathcal{L}_{\mathcal{D}}^{(3/4)\theta_i}(f) \leq l_{j+1}$ hence we know:

$$l_{j+1} \geq \mathcal{L}_{\mathcal{D}}^{(3/4)\theta_i}(f) \geq 8c_{3.2}\left(\frac{\ln(\theta_{i+1}^2 n/\ln(|\mathcal{H}|))\ln(|\mathcal{H}|)}{\theta_{i+1}^2 n} + \frac{\ln(e/\delta)}{n}\right)$$

$$\geq \frac{\ln(\theta_{i+1}^2 n/\ln(|\mathcal{H}|))\ln(|\mathcal{H}|)}{\theta_{i+1}^2 n} + \frac{\ln(e/\delta)}{n} \geq \frac{\ln(|\mathcal{H}|)}{\theta_{i+1}^2 n} \tag{22}$$

Firstly, since this is greater than $n^{-1}$, we know $j \neq 0$ and therefore $l_{j+1} = 2l_j$ and $l_{j+1} \leq 2\mathcal{L}_{\mathcal{D}}^{(3/4)\theta_i}(f)$. Applying this and (22) in (20) gives:

$$\mathcal{L}_{\mathbf{S}}^{\theta}(f) \leq \mathcal{L}_{\mathcal{D}}(f) + c_{3.1}\left(\sqrt{l_{j+1}\left(\frac{\ln(e/l_{j+1})\ln(|\mathcal{H}|)}{\theta_{i+1}^2 n} + \frac{\ln(e/\delta)}{n}\right)} + \frac{\ln(e/l_{j+1})\ln(|\mathcal{H}|)}{\theta_{i+1}^2 n} + \frac{\ln(e/\delta)}{n}\right)$$

$$\leq \mathcal{L}_{\mathcal{D}}^{(3/4)\theta_i}(f) + c_{3.1}\left(\sqrt{2\mathcal{L}_{\mathcal{D}}^{(3/4)\theta_i}(f)\left(\frac{\ln(e\theta_{i+1}^2 n/\ln(|\mathcal{H}|))\ln(|\mathcal{H}|)}{\theta_{i+1}^2 n} + \frac{\ln(e/\delta)}{n}\right)} + \frac{\ln(e\theta_{i+1}^2 n/\ln(|\mathcal{H}|))\ln(|\mathcal{H}|)}{\theta_{i+1}^2 n} + \frac{\ln(e/\delta)}{n}\right)$$

$$\leq \mathcal{L}_{\mathcal{D}}^{(3/4)\theta_i}(f) + c_{3.1}\left(\sqrt{2\mathcal{L}_{\mathcal{D}}^{(3/4)\theta_i}(f)\mathcal{L}_{\mathcal{D}}^{(3/4)\theta_i}(f)} + \mathcal{L}_{\mathcal{D}}^{(3/4)\theta_i}(f)\right) \leq 3c_{3.1}\mathcal{L}_{\mathcal{D}}^{(3/4)\theta_i}(f)$$

Hence $(3c_{3.1})^{-1}\mathcal{L}_{\mathbf{S}}^{\theta}(f) \leq \mathcal{L}_{\mathcal{D}}^{(3/4)\theta_i}(f) \leq l_{j+1}$, but more importantly $l_{j+1}^{-1} \leq 3c_{3.1}\mathcal{L}_{\mathbf{S}}^{\theta}(f)^{-1}$.

Additionally, applying (22) together with $\mathcal{L}_{\mathbf{S}}^{\theta}(f) \geq \mathcal{L}_{\mathbf{S}}^{\theta_i}(f)$ in (21) gives:

$$\mathcal{L}_{\mathbf{S}}^{\theta}(f) \geq \mathcal{L}_{\mathbf{S}}^{\theta_i}(f) \geq \frac{\mathcal{L}_{\mathcal{D}}^{(3/4)\theta_i}(f)}{2} - c_{3.2}\left(\frac{\ln(\theta_{i+1}^2 n/\ln(|\mathcal{H}|))\ln(|\mathcal{H}|)}{\theta_{i+1}^2 n} + \frac{\ln(e/\delta)}{n}\right)$$

$$\geq \frac{l_{j+1}}{4} - \frac{l_{j+1}}{8} = \frac{l_{j+1}}{8}$$

Hence $l_{j+1} \leq 8\mathcal{L}_{\mathbf{S}}^{\theta}(f)$, combining this with $l_{j+1}^{-1} \leq \theta_{i+1}^2 n/\ln(|\mathcal{H}|)$ and the previous conclusion $l_{j+1}^{-1} \leq 3c_{3.1}\mathcal{L}_{\mathbf{S}}^{\theta}(f)^{-1}$ and applying to (20) finally gives us:

$$\mathcal{L}_{\mathcal{D}}(f) \leq \mathcal{L}_{\mathbf{S}}^{\theta}(f) + c_{3.1}\left(\sqrt{l_{j+1}\left(\frac{\ln(e/l_{j+1})\ln(|\mathcal{H}|)}{\theta_{i+1}^2 n} + \frac{\ln(e/\delta)}{n}\right)} + \frac{\ln(e/l_{j+1})\ln(|\mathcal{H}|)}{\theta_{i+1}^2 n} + \frac{\ln(e/\delta)}{n}\right)$$

$$\leq \mathcal{L}_{\mathbf{S}}^{\theta}(f) + c_{3.1}\left(\sqrt{8\mathcal{L}_{\mathbf{S}}^{\theta}(f)\left(\frac{\ln(e3c_{3.1}/\mathcal{L}_{\mathbf{S}}^{\theta}(f))\ln(|\mathcal{H}|)}{\theta_{i+1}^2 n} + \frac{\ln(e/\delta)}{n}\right)} + \frac{\ln(e\theta_{i+1}^2 n/\ln(|\mathcal{H}|))\ln(|\mathcal{H}|)}{\theta_{i+1}^2 n} + \frac{\ln(e/\delta)}{n}\right)$$

which finishes the proof of the first part, since $\theta \leq \theta_{i+1}$

Now that we have established the implications of the events (9) and (10) holding simultaneously, we can continue by proving that this happens with probability $1 - \delta$ over $\mathbf{S} \sim \mathcal{D}^n$.

From Lemmas 3.1 and 3.2, we know (20) holds with probability $1 - \delta_{i,j}$, and (21) holds with probability $1 - \delta_j$. Let now

$$\delta_{i,j} = \left(\frac{\delta}{e}\right)^3 \exp\left(-\frac{\ln(e/l_{j+1})\ln(|\mathcal{H}|)}{\theta_{i+1}^2}\right) \qquad \text{and} \qquad \delta_i = \left(\frac{\delta}{e}\right)^3 \exp\left(-\frac{\ln(e\theta_{i+1}^2 n)\ln(|\mathcal{H}|)}{\theta_{i+1}^2}\right)$$

Then:

$$
\sum_{i=1}^{\log_2((c_\theta/e)\sqrt{n/\ln|\mathcal{H}|})} \sum_{j=0}^{\log_2(n)} \delta_{i,j} = \sum_{i=1}^{\log_2((c_\theta/e)\sqrt{n/\ln|\mathcal{H}|})} \sum_{j=0}^{\log_2(n)} \left(\frac{\delta}{e}\right)^3 \exp\left(-\frac{\ln(e/l_{j+1})\ln(|\mathcal{H}|)}{\theta_{i+1}^2}\right)
$$

$$
= \left(\frac{\delta}{e}\right)^3 \cdot \sum_{i=1}^{\log_2((c_\theta/e)\sqrt{n/\ln|\mathcal{H}|})} \sum_{j=0}^{\log_2(n)} \exp\left(-\frac{\ln(\frac{en}{2^j})\ln(|\mathcal{H}|)n}{2^{2i}}\right)
$$

$$
= \left(\frac{\delta}{e}\right)^3 \cdot \sum_{i=1}^{\log_2((c_\theta/e)\sqrt{n/\ln|\mathcal{H}|})} \sum_{j=0}^{\log_2(n)} \left(\frac{en}{2^j}\right)^{-\frac{n\ln(|\mathcal{H}|)}{2^{2i}}}
$$

summing the other direction, i.e substituting $i \leftarrow \log_2((c_\theta/e)\sqrt{n/\ln|\mathcal{H}|}) + 1 - i$ and $j \leftarrow \log_2(n) - j$, this sum equals:

$$
= \left(\frac{\delta}{e}\right)^3 \cdot \sum_{i=1}^{\log_2((c_\theta/e)\sqrt{n/\ln|\mathcal{H}|})} \sum_{j=0}^{\log_2(n)} \left(\frac{en}{2^{\log_2(n)-j}}\right)^{-\frac{n\ln(|\mathcal{H}|)}{2^{2-2i+2\log_2((c_\theta/e)\sqrt{n/\ln|\mathcal{H}|})}}}
$$

$$
= \left(\frac{\delta}{e}\right)^3 \cdot \sum_{i=1}^{\log_2((c_\theta/e)\sqrt{n/\ln|\mathcal{H}|})} \sum_{j=0}^{\log_2(n)} \left(e2^j\right)^{-\frac{\ln(|\mathcal{H}|)^2 e^2 2^{2i-2}}{c_\theta^2}}
$$

$$
\leq \left(\frac{\delta}{e}\right)^3 \cdot \sum_{i=1}^{\log_2((c_\theta/e)\sqrt{n/\ln|\mathcal{H}|})} \sum_{j=0}^{\log_2(n)} \left(e2^j\right)^{-\frac{\ln(|\mathcal{H}|)2^{2i-2}}{c_\theta^2}}
$$

$$
= \left(\frac{\delta}{e}\right)^3 \cdot \sum_{i=1}^{\log_2((c_\theta/e)\sqrt{n/\ln|\mathcal{H}|})} \sum_{j=0}^{\log_2(n)} \left(\frac{1}{e2^j}\right)^{\frac{\ln(|\mathcal{H}|)2^{2i-2}}{c_\theta^2}}
$$

$$
\leq \left(\frac{\delta}{e}\right)^3 \cdot \sum_{i=1}^{\log_2((c_\theta/e)\sqrt{n/\ln|\mathcal{H}|})} \sum_{j=0}^{\log_2(n)} \left(\frac{1}{e2^j}\right)^{2^{2i-2}}
$$

$$
\leq \left(\frac{\delta}{e}\right)^3 \cdot \sum_{i=1}^{\log_2((c_\theta/e)\sqrt{n/\ln|\mathcal{H}|})} 2\left(\frac{1}{e}\right)^{2^{2i-2}}
$$

$$
\leq \left(\frac{\delta}{e}\right)^3 \leq \delta/2
$$

i.e union bounding over all pairs $(\Theta_i, L_j)$ with $i = 1, \ldots, \log_2((c_\theta/e)\sqrt{n/\ln|\mathcal{H}|})$ and $j = 0, \ldots, \log_2(n)$, gives that (9) holds for all pairs with probability at least $1 - \delta/2$.

Doing the same for (10), gives:

$$
\sum_{i=1}^{\log_2((c_\theta/e)\sqrt{n/\ln|\mathcal{H}|})} \delta_i = \sum_{i=1}^{\log_2((c_\theta/e)\sqrt{n/\ln|\mathcal{H}|})} \left(\frac{\delta}{e}\right)^3 \exp\left(-\frac{\ln(e\theta_{i+1}^2 n)\ln(|\mathcal{H}|)}{\theta_{i+1}^2}\right)
$$

$$
\leq \left(\frac{\delta}{e}\right)^3 \sum_{i=1}^{\log_2((c_\theta/e)\sqrt{n/\ln|\mathcal{H}|})} \exp\left(-\ln(e\theta_{i+1}^2 n)\right)
$$

$$
= \left(\frac{\delta}{e}\right)^3 \sum_{i=1}^{\log_2((c_\theta/e)\sqrt{n/\ln|\mathcal{H}|})} \frac{1}{e\theta_{i+1}^2 n}
$$

$$
= \left(\frac{\delta}{e}\right)^3 \sum_{i=1}^{\log_2((c_\theta/e)\sqrt{n/\ln|\mathcal{H}|})} \frac{n}{e^3 \ln|\mathcal{H}| 2^{2i} n}
$$

$$
\leq \left(\frac{\delta}{e}\right)^3 \sum_{i=1}^{\log_2((c_\theta/e)\sqrt{n/\ln|\mathcal{H}|})} \frac{1}{e 2^{2i}}
$$

$$
\leq \left(\frac{\delta}{e}\right)^3 \leq \delta/2
$$

Finally remarking that:

$$
e/\delta_{i,j} \leq \left(\frac{e}{\delta}\right)^4 \exp\left(\frac{\ln(e/l_{j+1})\ln(|\mathcal{H}|)}{\theta_{i+1}^2}\right) \qquad \text{and} \qquad e/\delta_i \leq \left(\frac{e}{e}\right)^4 \exp\left(\frac{\ln(e\theta_{i+1}^2 n)\ln(|\mathcal{H}|)}{\theta_{i+1}^2}\right)
$$

implying that

$$
\frac{\ln(e/\delta_{i,j})}{n} \leq \frac{\ln\left((e/\delta)^4 \cdot \exp\left(\frac{\ln(e/l_{j+1})\ln(|\mathcal{H}|)}{\theta_{i+1}^2}\right)\right)}{n} = 4\frac{\ln(e/\delta)}{n} + \frac{\ln(e/l_{j+1})\ln(|\mathcal{H}|)}{\theta_{i+1}^2 n}
$$

and

$$
\frac{\ln(e/\delta_i)}{n} \leq \frac{\ln\left((e/\delta)^4 \exp\left(\frac{\ln(e\theta_{i+1}^2 n)\ln(|\mathcal{H}|)}{\theta_{i+1}^2}\right)\right)}{n} = 4\frac{\ln(e/\delta)}{n} + \frac{\ln(e\theta_{i+1}^2 n)\ln(|\mathcal{H}|)}{\theta_{i+1}^2 n}
$$

Hence in conclusion, we have proved that with probability $1 - \delta$ over the sample $\mathbf{S} \sim \mathcal{D}^n$ it holds for all $(\Theta_i, L_j)$, that:

$$
\sup_{\substack{f \in \mathcal{C}_\mathcal{H}(\Theta_i, L_j) \\ \theta \in \Theta_i}} |\mathcal{L}_\mathcal{D}(f) - \mathcal{L}_\mathbf{S}^\theta(f)| \leq c\left(\sqrt{l_{j+1}\left(\frac{\ln(e/l_{j+1})\ln(|\mathcal{H}|)}{\theta_{i+1}^2 n} + \frac{\ln(e/\delta_{i,j})}{n}\right)} + \frac{\ln(e/l_{j+1})\ln(|\mathcal{H}|)}{\theta_{i+1}^2 n} + \frac{\ln(e/\delta_{i,j})}{n}\right)
$$

$$
\leq c\left(\sqrt{l_{j+1}\left(2\frac{\ln(e/l_{j+1})\ln(|\mathcal{H}|)}{\theta_{i+1}^2 n} + 4\frac{\ln(e/\delta)}{n}\right)} + 2\frac{\ln(e/l_{j+1})\ln(|\mathcal{H}|)}{\theta_{i+1}^2 n} + 4\frac{\ln(e/\delta)}{n}\right)
$$

as well as for all $\Theta_i$, it holds that

$$
\forall f \in \mathcal{C}_\mathcal{H} : \qquad \frac{\mathcal{L}_\mathcal{D}^{(3/4)\theta_i}(f)}{2} - \mathcal{L}_\mathbf{S}^{\theta_i}(f) \leq c\left(\frac{\ln(\theta_{i+1}^2 n/\ln|\mathcal{H}|)\ln(|\mathcal{H}|)}{\theta_{i+1}^2 n} + \frac{\ln(e/\delta_i)}{n}\right)
$$

$$
\leq c\left(2\frac{\ln(\theta_{i+1}^2 n/\ln|\mathcal{H}|)\ln(|\mathcal{H}|)}{\theta_{i+1}^2 n} + 4\frac{\ln(e/\delta)}{n}\right)
$$

finishing the proof of the second part of the claim.

$\square$

**D.2. Replacement with $\phi$ and $\rho$**

**Restatement of Lemma 3.4.**

$$\underset{\mathbf{g}\sim\mathcal{Q}_f}{\mathbb{E}}\left[\underset{(\mathbf{x},\mathbf{y})\sim\mathcal{D}}{\mathbb{P}}[\mathbf{y}\mathbf{g}(\mathbf{x}) > \theta/2 \wedge \mathbf{y}f(\mathbf{x}) \leq 0]\right] \leq \underset{(\mathbf{x},\mathbf{y})\sim\mathcal{D}}{\mathbb{E}}[\phi(\mathbf{y}f(\mathbf{x}))]$$

$$\underset{\mathbf{g}\sim\mathcal{Q}_f}{\mathbb{E}}\left[\underset{(\mathbf{x},\mathbf{y})\sim S}{\mathbb{P}}[\mathbf{y}\mathbf{g}(\mathbf{x}) > \theta/2 \wedge \mathbf{y}f(\mathbf{x}) \leq \theta]\right] \geq \underset{(\mathbf{x},\mathbf{y})\sim S}{\mathbb{E}}[\phi(\mathbf{y}f(\mathbf{x}))]$$

$$\underset{\mathbf{g}\sim\mathcal{Q}_f}{\mathbb{E}}\left[\underset{(\mathbf{x},\mathbf{y})\sim S}{\mathbb{P}}[\mathbf{y}\mathbf{g}(\mathbf{x}) \leq \theta/2 \wedge \mathbf{y}f(\mathbf{x}) > \theta]\right] \leq \underset{(\mathbf{x},\mathbf{y})\sim S}{\mathbb{E}}[\rho(\mathbf{y}f(\mathbf{x}))]$$

$$\underset{\mathbf{g}\sim\mathcal{Q}_f}{\mathbb{E}}\left[\underset{(\mathbf{x},\mathbf{y})\sim\mathcal{D}}{\mathbb{P}}[\mathbf{y}\mathbf{g}(\mathbf{x}) \leq \theta/2 \wedge \mathbf{y}f(\mathbf{x}) > 0]\right] \geq \underset{(\mathbf{x},\mathbf{y})\sim\mathcal{D}}{\mathbb{E}}[\rho(\mathbf{y}f(\mathbf{x}))]$$

The proof uses the following monotonicity property, which we will prove after the Lemma

**Lemma D.2.** *for any $\lambda_1, \lambda_2$ such that $-1 < \lambda_1 \leq \lambda_2 < 1$:*

$$\underset{\mathbf{g}\sim\mathcal{Q}_f}{\mathbb{P}}[y\mathbf{g}(x) > \theta/2 \mid yf(x) = \lambda_1] \leq \underset{\mathbf{g}\sim\mathcal{Q}_f}{\mathbb{P}}[y\mathbf{g}(x) > \theta/2 \mid yf(x) = \lambda_2]$$

*Proof of Lemma 3.4.* Firstly we remark that we can swap what we take probability and expectation over:

$$\underset{\mathbf{g}\sim\mathcal{Q}_f}{\mathbb{E}}\left[\underset{(\mathbf{x},\mathbf{y})\sim\mathcal{D}}{\mathbb{P}}[\mathbf{y}\mathbf{g}(\mathbf{x}) > \theta/2 \wedge \mathbf{y}f(\mathbf{x}) \leq 0]\right] = \underset{(\mathbf{x},\mathbf{y})\sim\mathcal{D}}{\mathbb{E}}\left[\underset{\mathbf{g}\sim\mathcal{Q}_f}{\mathbb{P}}[\mathbf{y}\mathbf{g}(\mathbf{x}) > \theta/2 \wedge \mathbf{y}f(\mathbf{x}) \leq 0]\right]$$

Hence to prove the first inequality, we just need to prove $\mathbb{P}_{\mathbf{g}\sim\mathcal{Q}_f}[y\mathbf{g}(x) > \theta/2 \wedge yf(x) \leq 0] \leq \phi(yf(x))$ when $yf(x)$ is fixed (and similarly for the other three expressions). Lets start with the inequalities concerning $\phi$

By definition, $\phi(\lambda)$ depends on the value of $\lambda$

$$\phi(\lambda) = \begin{cases} \underset{\mathbf{g}\sim\mathcal{Q}_f}{\mathbb{P}}[y\mathbf{g}(x) > \theta/2 \mid yf(x) = \lambda] & -1 < \lambda \leq 0 \\ \dfrac{\theta - \lambda}{\theta} \underset{\mathbf{g}\sim\mathcal{Q}_f}{\mathbb{P}}[y\mathbf{g}(x) > \theta/2 \mid yf(x) = 0] & 0 < \lambda \leq \theta \\ 0 & \theta < \lambda \leq 1 \end{cases}$$

hence to prove the first inequality involving $\phi$, we split into cases:

- $-1 < yf(x) \leq 0$ implies:

$$\underset{\mathbf{g}\sim\mathcal{Q}_f}{\mathbb{P}}[y\mathbf{g}(x) > \theta/2 \wedge yf(x) \leq 0] = \underset{\mathbf{g}\sim\mathcal{Q}_f}{\mathbb{P}}[y\mathbf{g}(x) > \theta/2 \wedge yf(x) \leq 0 \mid -1 < yf(x) \leq 0]$$
$$= \underset{\mathbf{g}\sim\mathcal{Q}_f}{\mathbb{P}}[y\mathbf{g}(x) > \theta/2 \mid -1 < yf(x) \leq 0] = \phi(yf(x))$$

- if $0 < yf(x) \leq 1$, then $\mathbb{P}_{\mathbf{g}\sim\mathcal{Q}_f}[y\mathbf{g}(x) > \theta/2 \wedge yf(x) \leq 0] = 0 \leq \phi(yf(x))$

This in total gives the first of the four inequalities.

For the second inequality we need to prove $\mathbb{P}_{\mathbf{g}\sim\mathcal{Q}_f}[y\mathbf{g}(x) > \theta/2 \wedge yf(x) \leq \theta] \geq \phi(yf(x))$, we split up into cases:

- $-1 < yf(x) \leq 0$ implies $yf(x) \leq \theta$ and hence:

$$\underset{\mathbf{g}\sim\mathcal{Q}_f}{\mathbb{P}}[y\mathbf{g}(x) > \theta/2 \wedge yf(x) \leq \theta] = \underset{\mathbf{g}\sim\mathcal{Q}_f}{\mathbb{P}}[y\mathbf{g}(x) > \theta/2 \wedge yf(x) \leq \theta \mid -1 < yf(x) \leq 0]$$
$$= \underset{\mathbf{g}\sim\mathcal{Q}_f}{\mathbb{P}}[y\mathbf{g}(x) > \theta/2 \mid -1 < yf(x) \leq 0] = \phi(yf(x))$$

- if $0 < yf(x) \le \theta$, then by definition and monotonicity (Lemma D.2),

$$\phi(yf(x)) \le \mathop{\mathbb{P}}_{\mathbf{g} \sim \mathcal{Q}_f} [y\mathbf{g}(x) > \theta/2 \mid yf(x) = 0] \le \mathop{\mathbb{P}}_{\mathbf{g} \sim \mathcal{Q}_f} [y\mathbf{g}(x) > \theta/2 \mid 0 < yf(x) \le \theta]$$

$$= \mathop{\mathbb{P}}_{\mathbf{g} \sim \mathcal{Q}_f} [y\mathbf{g}(x) > \theta/2 \wedge yf(x) \le \theta \mid 0 < yf(x) \le \theta] = \mathop{\mathbb{P}}_{\mathbf{g} \sim \mathcal{Q}_f} [y\mathbf{g}(x) > \theta/2 \wedge yf(x) \le \theta]$$

- if $\theta < yf(x) \le 1$, then $\phi(yf(x)) = 0 = \mathbb{P}_{\mathbf{g} \sim \mathcal{Q}_f} [y\mathbf{g}(x) > \theta/2 \wedge yf(x) \le \theta]$

This in total gives the second of the four inequalities

The arguments for $\rho$ are similar. $\qquad\square$

Now to complete the argument, we need to prove Lemma D.2.

*Proof of Lemma D.2.* Firstly by Remark D.4:

$$yf(x) = \lambda_i \implies \mathop{\mathbb{P}}_{\mathbf{h} \sim \mathcal{D}_f} [y\mathbf{h}(x) = 1] = \frac{1}{2} + \frac{\lambda_i}{2}$$

Now let $\mathbf{T} \sim U(0,1)$ be uniformly distributed, let $f(x)y = \lambda_i$ and $\mathbf{h} \sim \mathcal{D}_f$, then

$$\mathbf{h}(x)y \stackrel{d}{=} 2 \cdot \mathbb{1}_{\{\mathbf{T} \le \frac{1}{2} + \frac{\lambda_i}{2}\}}(\mathbf{T}) - 1$$

And most importantly, since $\lambda_1 \le \lambda_2$, for all $T \in [0,1]$

$$\mathbb{1}_{\{T \le \frac{1}{2} + \frac{\lambda_1}{2}\}}(T) \le \mathbb{1}_{\{T \le \frac{1}{2} + \frac{\lambda_2}{2}\}}(T)$$

Meaning that:

$$\frac{1}{N} \sum_{i \in [N]} (2 \cdot \mathbb{1}_{\{T \le \frac{1}{2} + \frac{\lambda_1}{2}\}}(T) - 1)) > \theta/2 \implies \frac{1}{N} \sum_{i \in [N]} (2 \cdot \mathbb{1}_{\{T \le \frac{1}{2} + \frac{\lambda_2}{2}\}}(T) - 1)) > \theta/2$$

And hence, in probability

$$\mathop{\mathbb{P}}_{\mathbf{g} \sim \mathcal{Q}_f} [y\mathbf{g}(x) > \theta/2 \mid yf(x) = \lambda_1]$$

$$= \mathop{\mathbb{P}}_{\{\mathbf{h}_1, \dots, \mathbf{h}_N\} \sim \mathcal{D}_f^N} \left[ \frac{1}{N} \sum_{i \in [N]} y\mathbf{h}_i(x) > \theta/2 \;\middle|\; yf(x) = \lambda_1 \right]$$

$$= \mathop{\mathbb{P}}_{\{\mathbf{T}_1, \dots, \mathbf{T}_N\} \sim U(0,1)^N} \left[ \frac{1}{N} \sum_{i \in [N]} (2 \cdot \mathbb{1}_{\{\mathbf{T} \le \frac{1}{2} + \frac{\lambda_1}{2}\}}(\mathbf{T}) - 1)) > \theta/2 \right]$$

$$\le \mathop{\mathbb{P}}_{\{\mathbf{T}_1, \dots, \mathbf{T}_N\} \sim U(0,1)^N} \left[ \frac{1}{N} \sum_{i \in [N]} (2 \cdot \mathbb{1}_{\{\mathbf{T} \le \frac{1}{2} + \frac{\lambda_2}{2}\}}(\mathbf{T}) - 1)) > \theta/2 \right]$$

$$= \mathop{\mathbb{P}}_{\{\mathbf{h}_1, \dots, \mathbf{h}_N\} \sim \mathcal{D}_f^N} \left[ \frac{1}{N} \sum_{i \in [N]} y\mathbf{h}_i(x) > \theta/2 \;\middle|\; yf(x) = \lambda_2 \right]$$

$$= \mathop{\mathbb{P}}_{\mathbf{g} \sim \mathcal{Q}_f} [y\mathbf{g}(x) > \theta/2 \mid yf(x) = \lambda_2]$$

the wanted monotonicity holds $\qquad\square$

## D.3. Evaluating $\phi$ and $\rho$

**Lemma D.3.** *For all $f \in \mathcal{C}(\mathcal{H})$, $\eta, \lambda \in [-1, 1]$ it holds that:*

$$\mathbb{P}_{\mathbf{g} \sim \mathcal{Q}_f}[y\mathbf{g}(x) > \eta \mid yf(x) = \lambda] = \mathbb{P}_{\mathbf{H} \sim Binom(N, p_h)}[\mathbf{H} \geq k^*]$$

*where $p_h = \frac{1}{2} + \frac{\lambda}{2}$ and $k^* = \lfloor (\frac{\eta}{2} + \frac{1}{2})N \rfloor + 1$*

Firstly we argue

*Remark* D.4. $yf(x) = \lambda \implies \mathbb{P}_{\mathbf{h} \sim \mathcal{D}_f}[y\mathbf{h}(x) = 1] = \frac{1}{2} + \frac{\lambda}{2}$

*Proof.* Computing the expected margin of a base classifier in two ways gives the claim

$$\mathbb{E}_{\mathbf{h} \sim \mathcal{D}_f}[y\mathbf{h}(x)] = y \mathbb{E}_{\mathbf{h} \sim \mathcal{D}_f}[\mathbf{h}(x)] = y \sum_{h \in \mathcal{H}} \mathbb{P}_{\mathbf{h} \sim \mathcal{D}_f}[\mathbf{h} = h]h(x) = y \sum_{h \in \mathcal{H}} a_h h(x) = yf(x) = \lambda$$

$$\mathbb{E}_{\mathbf{h} \sim \mathcal{D}_f}[y\mathbf{h}(x)] = \mathbb{P}_{\mathbf{h} \sim \mathcal{D}_f}[y\mathbf{h}(x) = 1] - \mathbb{P}_{\mathbf{h} \sim \mathcal{D}_f}[y\mathbf{h}(x) = -1] = 2 \mathbb{P}_{\mathbf{h} \sim \mathcal{D}_f}[y\mathbf{h}(x) = 1] - 1$$

$\square$

*Proof.* Each $g \in \mathcal{C}_N$ sampled using $\mathcal{Q}_f$ is defined by $1/N \sum_{i \in [N]} h_i$, where the $h_i$'s are sampled from $\mathcal{H}$ using $\mathcal{D}_f$. Hence the margin of $g$ depends on the margins of the $h_i$'s, or equivalently on the correctness of their prediction on $(x, y)$.

$$yg(x) = \frac{1}{N} \sum_{i \in [N]} yh_i(x) = \frac{|\{\text{correct } h_i\}| - |\{\text{wrong } h_i\}|}{N}.$$

Therefore we have an equivalence of events, when $yf(x) = \lambda$:

$$yg(x) > \eta \iff |\{\text{correct } h_i\}| > \left(\frac{\eta}{2} + \frac{1}{2}\right)N \iff |\{\text{correct } h_i\}| \geq \left\lfloor \left(\frac{\eta}{2} + \frac{1}{2}\right)N \right\rfloor + 1$$

hence we have an equality of probabilities.

$$\mathbb{P}_{\mathbf{g} \sim \mathcal{Q}_f}[y\mathbf{g}(x) > \theta/2 \mid yf(x) = \lambda] = \mathbb{P}_{(\mathbf{h}_1, \ldots, \mathbf{h}_N) \sim \mathcal{D}_f^N}\left[|\{\text{correct } h_i\}| \geq \left\lfloor \left(\frac{1}{2} + \frac{\theta}{4}\right)N \right\rfloor + 1 \,\middle|\, yf(x) = \lambda\right]$$

Now by Remark D.4, the probability $p_{h_i}$ that $h_i$ is correct on $(x, y)$, is $p_h = \frac{1}{2} + \frac{\lambda}{2}$, for all $i \in [N]$.

Let $k^* = \lfloor \left(\frac{1}{2} + \frac{\theta}{4}\right) N \rfloor + 1$, then the above event corresponds to getting at least $k^*$ successes out of $N$ consecutive Bernoulli trials with success probability $p_h$. Which is exactly the statement of the lemma. $\square$

