# OpenReview forum: "Tight Margin-Based Generalization Bounds for Voting Classifiers over Finite Hypothesis Sets"
_ICML.cc/2026/Conference — ICML 2026 regular_

### Official Review · Reviewer_xLDf · 2026-02-25

**Soundness:** 4
**Presentation:** 3
**Significance:** 4
**Originality:** 3
**Overall Recommendation:** 5
**Confidence:** 5

**Summary:**

This paper proves a margin-based, asymptotically tight generalization bound for voting classifiers.

**Compliance With Llm Reviewing Policy:**

Affirmed.

**Final Justification:**

There is no rebuttal, so I maintain my original score.

**Key Questions For Authors:**

N/A

**Limitations:**

yes

**Strengths And Weaknesses:**

Overall, this paper tackles a long-standing question. Although the results are confined to the finite $\mathcal{H}$ setting, they sharpen our understanding of what margin-based explanations of generalization can and cannot achieve, and provide a clear benchmark for subsequent work. The arguments are sound, and the presentation is good, I enjoy reading this paper.

---

### Official Review · Reviewer_FHza · 2026-03-10

**Soundness:** 4
**Presentation:** 4
**Significance:** 3
**Originality:** 3
**Overall Recommendation:** 4
**Confidence:** 4

**Summary:**

The paper extends generalisation analysis techniques to obtain tight upper and lower bounds for margin based bounds on the generalisation of voting classifiers over finite base classes following the line of work initiated by Schapire et al (1998). The work provides details of the novel approaches introduced giving clear descriptions of their origins in other works. The proofs are non-trivial and clearly presented as is the implication for the open questions in the literature.

**Compliance With Llm Reviewing Policy:**

Affirmed.

**Key Questions For Authors:**

Are there any algorithmic implications or insights that arise from the new proof approaches?

**Limitations:**

As mentioned above no experimental confirmation is included or discussed.

**Strengths And Weaknesses:**

Strengths:
* rigorous analysis providing matching upper and lower bounds for margin based generalisation bounds on voting classifiers over finite sets of weak learners.
* very clear presentation of context, proof intuitions and proof details.
* extension and addition of new theoretical techniques that may assist in further analyses
Weaknesses:
* no experimental confirmation of the results - how significant was the gap that has been closed?
* no algorithmic implications of the work are discussed.

---

> ### Author Rebuttal · Authors · 2026-03-30
>
> Thank you for your review. Regarding your question on algorithmic implications. Probably not. Our results should mostly be appreciated for giving the final and complete theoretical understanding of generalisation of voting classifiers.

---

### Official Review · Reviewer_v2sh · 2026-03-12

**Soundness:** 4
**Presentation:** 4
**Significance:** 4
**Originality:** 4
**Overall Recommendation:** 5
**Confidence:** 4

**Summary:**

This paper investigates margin-based generalization bounds for voting classifiers (ensembles) over finite base hypothesis sets. While the foundational framework of margin theory has long been used to explain the success of algorithms like AdaBoost, an asymptotic gap of $\sqrt{\ln(n)/\ln(e/\mathcal{L}_{S}^{\theta}(f))}$ remained between the tightest known upper bounds (Gao & Zhou, 2013) and lower bounds (Grønlund et al., 2020). This paper definitively closes that gap. By extending a recent framework by Larsen & Schalburg (2025) to bound the discretization cost using Rademacher complexity, the authors prove an asymptotically tight generalization upper bound that perfectly matches the lower bound across the tradeoff of parameters (hypothesis set size, margin, fraction of margin errors, sample size, and failure probability).

**Compliance With Llm Reviewing Policy:**

Affirmed.

**Key Questions For Authors:**

.

**Limitations:**

yes

**Strengths And Weaknesses:**

Strength: The paper closes a basic theoretical problem
Weaknesses: The improvement over previous SOT is quite marginal.

---

### Official Review · Reviewer_4fDV · 2026-03-13

**Soundness:** 4
**Presentation:** 4
**Significance:** 2
**Originality:** 1
**Overall Recommendation:** 3
**Confidence:** 4

**Summary:**

The paper proves the first asymptotically tight margin-based generalization bound for voting classifiers over finite hypothesis sets. Prior work established upper bounds relating the generalization error to the empirical margin distribution and hypothesis class size, while later results showed nearly matching lower bounds but left a logarithmic gap. This work closes that gap by deriving an improved upper bound that matches known lower bounds across the relevant parameter regimes. The proof refines the classic discretization framework of Schapire et al. (1998) and incorporates techniques from recent work on large-margin halfspaces by obtaining a sharper analysis of the discretization error.

**Compliance With Llm Reviewing Policy:**

Affirmed.

**Final Justification:**

I have justified my choice in the rebuttal acknowledgement.

**Key Questions For Authors:**

## Questions for the Authors

1. The proof builds on the discretization framework of Schapire et al. (1998) and appears to rely heavily on techniques introduced in recent work on large-margin halfspaces (e.g., Larsen & Schalburg, 2025). The improvement over prior bounds appears to come from a more careful treatment of terms that earlier analyses discarded. Could the authors clarify more precisely what elements of the argument are fundamentally new in this paper, as opposed to direct adaptations of these prior techniques? Additionally, could the authors provide intuition on why this refined analysis is essential for closing the logarithmic gap, and why it was not achievable within earlier proof approaches?

2.  The paper focuses on voting classifiers over **finite hypothesis classes**. Do the techniques introduced here extend to the more general setting of hypothesis classes with finite VC dimension, or are there inherent obstacles that prevent such an extension? Given that the result tightens the theoretical understanding of margin-based generalization, do the authors expect these techniques to have implications for the analysis or design of boosting algorithms beyond AdaBoost (e.g., algorithms that explicitly optimize margins)?

5. While the bound is asymptotically tight, could the authors comment on whether the improvement meaningfully changes the interpretation of margin-based generalization in practical regimes (e.g., realistic sample sizes or hypothesis class sizes)?

**Strengths And Weaknesses:**

## Strengths

1. The paper appears mathematically solid and builds on a well-established line of work on margin-based generalization bounds for voting classifiers. The proof carefully extends the classical discretization framework of Schapire et al. (1998) and integrates more recent techniques (e.g., Rademacher-based analysis) to obtain tighter bounds. The paper is also very well written.

2. The main contribution is an improved upper bound that asymptotically matches existing lower bounds for voting classifiers over finite hypothesis sets, thereby closing a logarithmic gap that persisted in prior work.

## Weaknesses

1. Much of the conceptual framework underlying the result (both the lower bounds and the proof machinery) derives from prior work on margin-based generalization bounds and recent techniques for analyzing large-margin classifiers. The main contribution is to refine existing analyses by tightening logarithmic factors.

3. The result addresses a fairly specialized question within margin-based generalization theory for boosting. As a result, the paper’s impact may primarily appeal to a relatively small subset of the learning theory community rather than a broader ICML audience. Given that the contribution mainly closes a technical gap in a specialized theoretical line of work, the paper may be better suited to theory-focused venues (e.g., ALT or COLT) rather than a broad ML venue like ICML.

---

> ### Author Rebuttal · Authors · 2026-03-30
>
> Thank you for your review. Regarding your questions:
>
> 1. We would first like to point to our section giving the Proof Overview and Our Key Improvements where we try to discuss how the original framework of Schapire et al. is inadequate. Now compared to Larsen and Schalburg, indeed we follow their overall framework. However, everything needs to be analysed carefully and new crucial ideas are needed for the whole proof. The only thing their framework tells us, is to look at the equations in (6). From thereon, analysing for instance the Lipschitz constant of the phi function uses several intricate properties of voting classifiers and binomial distributions and has no overlap with the prior proof of Larsen and Schalburg. We believe the right comparison would be similar to saying that Rademacher complexity has been used in many different applications, but each application still needs new and original ideas and insights.
> Why these new techniques are needed is a little more difficult to say exactly. Perhaps the clearest answer is that using the prior approach by Schaphire et al., the best one can hope for is the bound just above "Out Key Improvements", and choosing the optimal N gives a suboptimal bound compared to ours. In particular, the terms they are ignoring in their analysis just costs too much to ignore to get optimal bounds.
>
> 2. This is more subtle and the answer is both a yes and no. First, the current best bounds for finite VC-dimension are due to Hogsgaard and Larsen, COLT'25. Their analysis does not go via a randomized discretization and as such does not really fit into the framework of Larsen and Schalburg. Their bound has the same logarithmic sub optimality factor as we remove in this work. However, if one takes the previous randomized discretisation proof by Schaphire et al., then in the finite VC-dimension case, it is several log factors suboptimal compared to the bound by Hogsgaard and Larsen. Thus if we tried, we would improve one log factor, but lose another. That is why we focussed solely on the finite case where we get completely optimal bounds.
>
> 3. Probably not much. We believe our contribution should be mostly seen as the final piece in a complete theoretical understanding of generalisation of voting classifiers, not as a major practical breakthrough. We believe such theoretical contributions are exciting nonetheless, as they finish off long standing theoretical questions.

---

> > ### Author Rebuttal · Reviewer_4fDV · 2026-04-02
> >
> > I believe that while the results obtained are significant (and it would be a disservice to claim otherwise), this paper should be sent to a venue where the theory of the paper is an end goal in itself. While ICML does indeed serve as a home to theory papers, I do not think that the results or techniques in this paper are exciting to a broad ML audience, and the author responses have failed to convince me otherwise. I would not recommend acceptance for this paper.

---

### Decision · Program_Chairs · 2026-04-30

**Decision:**

Accept (regular)

**Comment:**

The paper closes a long-standing logarithmic gap between upper and lower bounds for voting classifiers.

Strength:
- Closing a long-standing gap between upper and lower bounds.
- Clear presentation, mathematically solid result.

Weaknesses:
- Relatively minor technical innovation compared to prior work.
- While theoretically interesting, the results have no practical or algorithmic implications. (Or, at least, no such implications have been raised in the submission or the rebuttal.)

The reviewers were split in their weighting of strength against weaknesses. I recommend acceptance based on the theoretical interest in closing the gap between upper and lower bounds.